# Fast Reconstruction of Mixtures of Bernoulli Product Distributions

Sanyam Agarwal [1]   Pranjal Dutta [2]   Markus Bläser [1]

## Abstract

Mixtures of Bernoulli product distributions are a simple and widely used latent-variable model, with applications in e.g. recommendation systems, crowdsourcing, and medical data analysis. We consider the problem of reconstructing the mixture parameters from oracle access to its probability generating polynomial (PGP), for instance represented by a probabilistic generating circuit (PGC). We show that the parameters are uniquely identifiable for almost all mixtures, and give a randomized algorithm that exactly recovers the mixture weights and component marginals for mixtures of $r$ Bernoulli product distributions over $n$ variables using only $O(nr^2)$ oracle queries. The algorithm repeatedly applies restrictions to $O(r)$ variables, extracts low-degree coefficients, and then recovers the parameters using a moment-based tensor decomposition. To the best of our knowledge, this is the *first* exact reconstruction algorithm in this PGP oracle model with query complexity linear in $n$ and polynomial in $r$.

## 1. Introduction

Mixtures of product distributions are a simple but surprisingly expressive latent-variable model for unsupervised learning. Each component is a product distribution, and mixing induces dependencies between coordinates through an unobserved label. This model has been used across a range of applications and also serves as a convenient testbed for developing and analyzing learning algorithms. In this work we focus on the computational question of how efficiently such mixtures can be reconstructed when the dimension $n$ is large.

**Guiding Question.** *Can mixtures of product distributions be reconstructed with only $O(n)$ oracle queries?*

More concretely, suppose an unknown distribution over $\{0,1\}^n$ is a mixture of $r$ product distributions. Is there an algorithm whose query complexity is essentially linear in $n$ (assuming $r$ is modest compared to $n$), while still efficiently recovering all mixture parameters?

Mixtures of product distributions are among the most classical and widely studied latent-variable models. Each mixture component captures coordinate-wise independence, while the mixture introduces correlations through an unobserved (hidden) variable. Beyond their foundational role in probability and statistics, such models arise naturally in many modern applications. For example, they appear in image recognition, population genetics, recommendation systems, medical imaging, and crowdsourcing, among others (Pritchard et al., 2000; Juan & Vidal, 2004; Tomozei & Massoulié, 2014; Dawid & Skene, 1979). In fact, it is known that *any* boolean distribution can be represented as a mixture of Bernoulli product components (Grim, 2006).

In this work, we revisit the learning problem from an algebraic/symbolic point of view. Instead of relying solely on sample access, we consider a *symbolic* setting in which the distribution is presented through its *probability generating polynomial* [1] (PGP) in a compact representation. This access allows us to compute low-degree multilinear coefficients efficiently (equivalently, low-order moments), and enables a reconstruction strategy that is explicitly coefficient-driven. This perspective connects distribution learning with polynomial reconstruction and suggests new routes to fast and explicit algorithms in structured access models. A potential application of such a scenario is the case when we have already learned a probabilistic generating circuit (PGC) (Zhang et al., 2021). While such a representation supports tractable marginalization, we cannot simply read off the parameters of the mixture from the PGC, since it might store the PGP in a different, more efficient way, for instance, using DPPs as subcircuits. Our work extends the tractable operations that are available for PGC, in the spirit of Vergari et al. (2021), by adding parameter reconstruction with linear query complexity to the toolbox. This is essential as recov-

---

[1] Department of Computer Science, Universität des Saarlandes, Saarland Informatics Campus, Saarbrücken, Germany [2] College of Computing and Data Science, Nanyang Technological University (NTU), Singapore. Correspondence to: Sanyam Agarwal <agarwal@cs.uni-saarland.de>.

*Proceedings of the 43rd International Conference on Machine Learning*, Seoul, South Korea. PMLR 306, 2026. Copyright 2026 by the author(s).

[1] Since the joint distribution has finite support, PGP is the same as the probability generating function (PGF).

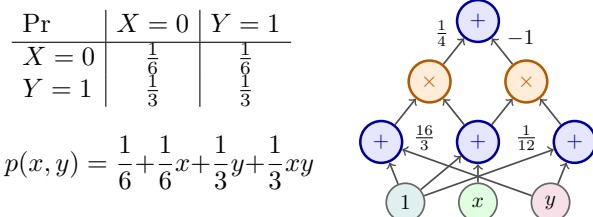

| Pr | $X = 0$ | $Y = 1$ |
|---|---|---|
| $X = 0$ | $\frac{1}{6}$ | $\frac{1}{6}$ |
| $Y = 1$ | $\frac{1}{3}$ | $\frac{1}{3}$ |

$$p(x,y) = \frac{1}{6} + \frac{1}{6}x + \frac{1}{3}y + \frac{1}{3}xy$$

*Figure 1.* The tabular distribution represented as both: A probability generating polynomial $p$, and the probabilistic generating circuit.

ery of parameters enables us to answer all sorts of inference queries helping us reason about the data.

**Model.** We study mixtures of Bernoulli product distributions. An $r$-component mixture on variables $x_1, \ldots, x_n$ can be written via its PGP as

$$f(x_1, \ldots, x_n) = \sum_{i=1}^{r} \mu_i \prod_{j=1}^{n} \big((1 - p_{ij})x_j + p_{ij}\big). \quad (1)$$

Here for all $i \in [r]$, $\mu_i \geq 0$ and $\sum_{i=1}^{r} \mu_i = 1$. Further, $p_{ij} \in [0, 1]$ for all $i \in [r], j \in [n]$, and denote the marginal probability of $x_j = 1$ in component $i$. For algebraic convenience, we will often work with the equivalent re-parameterization $(1 - p_{ij})x_j + p_{ij} = 1 + t_{ij}x_j$. We can formalize the above PGP-access model via *probabilistic generating circuits (PGCs)*, which compute the polynomial $f$ compactly. We are allowed to query the circuit beyond $\{0, 1\}^n$.

**Why probabilistic generating circuits?** A convenient way to represent a high-dimensional distribution over binary variables is through its *probability generating polynomial*, whose coefficients encode all joint probabilities, as shown in Figure 1. While this polynomial has exponentially many coefficients, recent work has shown that many structured distributions admit a compact representation as a *probabilistic generating circuit (PGC)*: a directed acyclic computation graph that evaluates the generating polynomial efficiently (Zhang et al., 2021; Harviainen et al., 2023; Agarwal & Bläser, 2024).

PGCs form a tractable probabilistic model class while also being expressive enough to subsume several classical tractable families including decomposable probabilistic circuits (PCs) and determinantal point processes (DPPs) (Zhang et al., 2021). PGCs sit within a growing view of tractable probabilistic models as succinct representations of multilinear polynomial semantics (Broadrick et al., 2025), where different polynomial encodings (network, likelihood, generating, etc.) can often be related by efficient transformations (Broadrick et al., 2024). Hence, our results transfer to other means of access to the distribution probabilities, for instance, through a probabilistic circuit (PC) (Poon & Domingos, 2011; Choi et al., 2020). Furthermore, there

has been substantial recent progress on improving both the learnability and scalability of such circuits, and practical algorithms for learning them from data are now available (Liu et al., 2024a; Gala et al., 2024; Zhang et al., 2025). A key advantage of these models is that they support exact inference queries efficiently, which makes them particularly attractive in applications where reliable probabilistic reasoning is important. Not only that, they have also been used in control and alignment of deep generative models (Liu et al., 2024b; Zhang et al., 2024) among other applications (Saad et al., 2021; Wedenig et al., 2024).

### 1.1. Our results

As our main result, we give an algorithm for recovering the parameters of a mixture of Bernoulli product distributions, as given by Equation 1. We assume that we have a query access to the PGP representing the distribution, for instance via PGCs. Our algorithm is designed for the "generic" setting. Gyllenberg et al. (1994) showed that mixtures of Bernoulli product distributions are nontrivially non-identifiable, i.e., there exists more than one set of parameter tuples (ignoring permutations of mixture components or coincident component distributions) that agree with the observed data. However, Allman et al. (2009) showed that the set of these non-identifiable parameters can be characterized through algebraic conditions and form a set of measure zero within the parameter space. (In algebraic geometry, this is called a Zariski-closed set, see Vakil, 2025.) Hence, we can "almost always" identify the parameters of models originating in practice.

*Remark* 1.1. In the generic setting, we assume that $p_{ij} \neq 0, 1$ for all $i \in [r]$ and $j \in [m]$ in Equation 1. This is a natural non-degeneracy assumption for mixtures of Bernoulli distributions. Formally, the set of parameters for which $p_{ij} \in \{0, 1\}$ for some $i, j$ is given by a set of algebraic conditions, namely $(p_{ij} - 1)p_{ij} = 0$.

To the best of our knowledge, we provide the first algorithm for reconstruction of parameters of a mixture of Bernoulli product distributions such that the required number of queries is *linear* in the number of variables and *quadratic* in the number of mixtures $r$, assuming the aforementioned genericity conditions. In addition, the algorithm recovers the parameters of the distribution given by the PGP *exactly*. Secondly, while inference queries like marginalization are known to be tractable for PGCs (Zhang et al., 2021), we *expand the toolbox of efficient operations* on PGCs (equivalently, probabilistic circuits), by demonstrating efficient parameter recovery in this setting. We now state our main result.

**Theorem 1.2.** *Given oracle access to a PGP representing mixture of Bernoulli product distributions as shown in (1), we give an algorithm that reconstructs the exact parameters in $O(nr^2)$ queries in the generic setting.*

*Remark* 1.3. If we have oracle access to the PGP the algorithm has a runtime of $O\Big(nr^{4+\delta}(\log r + \log \log \frac{1}{\epsilon}) + nr^{2+\omega}(\log r + \log \frac{1}{\epsilon})\Big)$ for small $\delta, \epsilon > 0$. In general, the running time also includes an *additive* factor of $O(nsr^2)$ where $s$ is the size of the representation of the PGP.

*Remark* 1.4. If the parameters were arbitrary irrationals, then no practical algorithm could hope to recover them with infinite precision. Hence, we assume that the parameters of the underlying distribution are specified up to some fixed precision $\epsilon$, i.e., for any parameters $a, b$, $|a - b| \leq \epsilon = 2^{-\gamma}$. The oracle is robust to precision losses due to marginalization, i.e., for a query with $n - k$ random variables set to 1 in (1), the output is a value with precision $2^{-k\gamma + \log r}$ or $r + k \log \frac{1}{\epsilon}$ bits.

A key technical ingredient—of independent interest—is an algebraic method to recover missing entries from off-diagonal moments, avoiding matrix-completion style routines (see Proposition E.1).

*Remark* 1.5. We only assume query access to the PGP/PGC and can't inspect the internal structure of the circuit. To our knowledge, no known algorithm using the circuit structure would improve the running time of our algorithm, since a PGC can store the PGP in a completely different way than the standard sum of products representation.

*Remark* 1.6. Determining the fan-in $r$ from black-box access to $f$ amounts to recovering $r$ as a function of the hidden parameters (e.g., via the Jacobian with respect to $\mu_i$ and $t_{i,j}$). In the worst case, this task is computationally hard: even deciding whether $f$ admits such a representation with fan-in at most $r$ is known to be NP-hard (as it subsumes tensor-rank), as shown in Håstad (1990). These lower bounds rely on carefully constructed instances and do not reflect the generic regime we study. Indeed, as a corollary of our results, we can compute $r$ in polynomial time (for $r \leq (n - 1)/4$, in the generic setting) by enumerating candidate values of $r$, running our reconstruction algorithm, and certifying the output via polynomial identity testing by comparing values up to $\log r + n \log \frac{1}{\epsilon}$ bits.

**Genericity.** We prove reconstruction in a distribution-free *generic* sense: the relevant Jacobian (or, equivalently, an appropriate collection of its maximal minors) is not identically zero as a polynomial in the parameters, so the set of parameters where our identifiability condition fails lies in a proper algebraic variety. Equivalently, under any continuous parameter distribution, this non-degeneracy condition holds with probability 1. Importantly, this argument does not constrain the mixture weights $\mu_i$.

## 1.2. Related works

**Learning mixtures of product distributions.** Learning such mixtures goes back to the classical EM algorithm of Dempster et al. (1977), and early complexity-theoretic investigations by Kearns et al. (1994) and Dasgupta (1999). Efficient PAC-style algorithms for mixtures of discrete distributions were developed in Feldman et al. (2008), with subsequent improvements in more structured regimes, e.g., under separation assumptions (Chaudhuri & Rao, 2008), and via faster spectral/tensor methods (Chen & Moitra, 2019). Much of this literature focuses on recovering a distribution close to the truth rather than the exact parameters, and typically runs in time $n^{\text{poly}(r)}$, which quickly becomes impractical as $r$ grows.

More recently, there has been progress on the *identification* problem of recovering a model close enough to the true model in the parameter space under additional assumptions. Gordon et al. (2021) gave a $2^{O(r^2)}n^{O(r)}$ algorithm for learning mixtures of $r$ products from approximate multilinear moments under separability conditions, further refined in Gordon et al. (2024). In their work, they assume that the moments are given to them through an oracle access. In particular, under certain "separability conditions" they even give a $n2^{O(r^2)}$ running time. While this is linear in $n$, the algorithm gets impractical even for moderate $r$. In a different direction, Jain & Oh (2014) obtained a polynomial-time algorithm with running time $\text{poly}(n, r)$ under a structured setting where the components satisfy certain well-conditioning assumptions. While polynomial, the dependence on $n$ is far from linear, and the approach relies on nontrivial linear-algebraic subroutines.

Algorithmically, a recurring theme is to exploit low-order moments (often second- and third-order) and perform a recovery step from these statistics, as in the general method-of-moments framework for latent-variable models (Anandkumar et al., 2014) using tensor decomposition tools such as the tensor power method (De Lathauwer et al., 2000). In the sample-based setting, moment estimation incurs sampling error; moreover, for Bernoulli product mixtures, Jain & Oh (2014) show that off-diagonal moment information alone does not directly provide the diagonal statistics needed for parameter recovery, motivating additional procedures such as matrix completion and least-squares based recovery.

**A circuit reconstruction viewpoint.** A second, complementary perspective comes from algebraic complexity theory. The probability generating polynomial of a mixture of Bernoulli product distributions admits a depth-3 sum-of-products representation (as in (1)), with top fan-in equal to the number of mixture components $r$. This links our setting to the literature on reconstructing structured polynomials from oracle (black-box) access. In the *worst case*, however, reconstruction for general depth-3 circuits is known to be highly expensive even for constant $r$: for example, Bhargava et al. (2020, Theorem 1.6) gives a randomized algorithm with running time $n^{r^{r^{r^{10}}}}$ for depth-3 multilinear

circuits. Subsequent results obtain polynomial-time guarantees in certain restricted regimes (Kayal & Saha, 2019), and quasipolynomial-time algorithms for constant $r$ in greater generality (Saraf et al., 2025), but these bounds remain far from practical, even for moderately small fan-in $r$.

### 1.3. Our perspective and comparison

Our work sits at the intersection of two viewpoints—structured distribution learning and symbolic/algebraic reconstruction—but in a different access model: we assume the mixture is given through its *probability generating polynomial* in a compact representation, formalized as a PGP oracle or via PGCs (Zhang et al., 2021; Broadrick et al., 2024; Agarwal & Bläser, 2024) or similar tractable models. An application scenario is the case where we have already learned the PGC and now want to reconstruct the parameters, which one cannot read off from the circuit, since the learned circuits can have a structure that is different from the standard encoding as a sum of products, In particular, PGCs are even allowed to use negative weights, or can have determinantal circuits as subroutines (Zhang et al., 2021). Compared to worst-case circuit reconstruction, we benefit from working in a generic identifiability regime, which is a reasonable scenario for machine learning applications (Allman et al., 2009). Second, the PGP/PGC representation provides efficient symbolic access to low-degree coefficients. Once the PGC is learned, we have symbolic access to the PGP of the learned distribution. In this way, we work in an inference-oracle (noiseless) model: rather than receiving random samples, the learner can query the underlying generating object to obtain exact low-degree coefficients / moment information. This aligns more closely with classical query learning (Angluin, 1988) and with mixture-learning works that assume oracle access to the target distribution.

We utilize these low-degree multilinear coefficients (equivalently, lower-order moments) in two ways. First, unlike the standard sample-based setting, we can obtain the relevant moments *exactly* from the representation, eliminating sampling error in the reconstruction pipeline. Second, we show that (in a generic sense) projections to a small number of variables preserve the underlying mixture/circuit structure, allowing us to learn the model on low-dimensional restrictions and stitch together the full solution across all $n$ variables. In the same spirit as Jain & Oh (2014), we address the issue of recovering diagonal information from off-diagonal statistics. In the generic setting, this step admits a particularly efficient algebraic solution, avoiding the need for matrix completion or least-squares style procedures allowing us to prove Theorem 1.2.

**Comparison with other linear time algorithms.** (Rabani et al., 2014) study mixtures of *arbitrary (unstructured)* distributions over a discrete domain $[N]$, where $N$ is the *domain*

*size* (e.g., vocabulary size), and their positive results rely on *multi-snapshot ("aperture")* samples (several i.i.d. draws from the *same* hidden component). In our setting, a single observation lies in $\{0, 1\}^n$, so the ambient domain has size $2^n$; thus, mapping our problem into their framework would correspond to $N = 2^n$, making their near-linear dependence on $N$ exponential in $n$ in our case.

Gordon et al. (2021; 2024) consider *source identification* for mixtures of $r$ product distributions on $n$ observable bits from conventional i.i.d. samples, taking as input empirical estimates of multilinear moments $g(S) = \mathbb{E}[X_S]$ up to order $|S| \leq 3r - 3$. Their reported (post-sampling) runtime is, in general, $2^{O(r^2)} \cdot n^{O(r)}$ arithmetic operations. Their strongest linear-in-$n$ runtime regime requires a quantitative $\zeta$-separation condition on the observables, and even then the runtime includes a factor $2^{O(r^2)}$, which renders the algorithm impractical even for moderate $r$.

## 2. Proof idea and algorithm overview

We are given oracle access to the probability generating polynomial (PGP) $f(x_1, \ldots, x_n) = \sum_{i=1}^r \mu_i \prod_{j=1}^n \left((1 - p_{ij})x_j + p_{ij}\right)$, equivalently represented by a probabilistic generating circuit (PGC). Our goal is to exactly recover the weights $\{\mu_i\}_{i=1}^r$ and marginals $\{p_{ij}\}_{i=1,j=1}^{i=r,j=n}$ using as few oracle queries as possible.

**Preprocessing and genericity.** We first apply a preprocessing step to remove variables that do not appear in the PGP and to factor out any linear terms that are common to all mixture components, both of which can be detected using only $O(n)$ oracle queries; see Appendix A. Hereon, our guarantees hold under mild genericity assumptions (holding for almost all parameter choices) on the parameters $p_{ij}$, which rule out degenerate cases such as identical mixture components or parameters lying on a proper algebraic sub-variety. In particular, our algorithm doesn't require any restrictions on the mixture weights $\mu_i$.

A key challenge now is that we cannot work with all $n$ variables at once. Instead, our algorithm repeatedly restricts the PGP to only $m = O(r)$ variables (keeping the rest fixed), solves the reconstruction problem in this low-dimensional projection, and then stitches the solutions across blocks to recover all $n$ coordinates. This approach requires resolving two conceptual issues:

**(1) Small restrictions must preserve fan-in $r$.** When we restrict to a subset of variables, it is not *a priori* clear that the resulting polynomial still corresponds to a mixture of $r$ distinct product components; in principle, different components could collapse and the effective fan-in could drop below $r$. To rule this out, we study the coefficient map induced by a restriction and show that, for almost all parameter choices, the restricted PGP cannot be represented as a

mixture of fewer than $r$ products. Technically, we formalize this via a Jacobian rank separation argument: even though the Jacobian is *not full rank* in our setting, its rank is provably larger for an $r$-mixture than for an $(r-1)$-mixture, which certifies that the restriction preserves the mixture size generically, see Appendix C.

**(2) Identifiability and exact recovery from low-order coefficients.** Once a restriction preserves fan-in $r$, we must *recover* the parameters from that restricted instance. Oracle access to the PGP allows us to extract low-degree multilinear coefficients exactly via interpolation using only polynomially many queries in $m$ (hence polynomial in $r$). These low-degree coefficients can be organized as second- and third-order moment objects of the restricted model, leading to a structured low-rank tensor of the form $T = \sum_{i=1}^{r} \mu_i u_i^{\otimes 3}$. We then apply a moment-based tensor decomposition routine (whitening followed by a tensor power method) to recover the vectors $u_i$ and weights $\mu_i$ (up to permutation); see Section 4.2 and Algorithm 2. To justify correctness, we show in Appendix G that the resulting rank-$r$ tensor decomposition is generically unique (up to permutation and scaling) via Kruskal's theorem, ensuring that the recovered components correspond to the true mixture parameters.

A technical obstacle in prior spectral/sample-based approaches is that the required low-rank moment matrix/tensor *cannot* be read off directly from observed moments because certain diagonal blocks are missing; these are typically handled via matrix completion or least-squares optimization. In contrast, in our generic setting we recover the missing diagonal statistics *algebraically* by exploiting the fact that the relevant moment matrix has rank at most $r$, so every $(r+1) \times (r+1)$ minor vanishes. Choosing a minor that contains exactly one unknown diagonal entry gives a single linear equation in that entry, which we can solve in closed form without any numerical optimization. See Section 4.1 and Algorithm 1.

**Stitching across blocks.** A single restriction to $m = O(r)$ variables only reveals the parameters on those coordinates, and even within one block the recovered components are a priori defined only up to permutation (and scaling). However, it can be shown that the relevant rank-$r$ tensor decomposition is generically unique, so the restricted parameters are well-defined up to a permutation of components. You can find the proof in Appendix G. To recover the full matrix $(p_{ij})$ over all $n$ variables, we partition the variables into blocks of size $m = O(r)$ and reconstruct the restricted parameters blockwise. Since each block recovery is only defined up to a permutation of components, we align these permutations globally using either the mixture weights (when distinct) or by adding a small number of anchor variables to later blocks to avoid collisions (see Appendix H for the full technical argument). This yields exact global parameter

recovery with query complexity linear in $n$ and per-block cost polynomial in $r$.

## 3. Small restrictions preserve mixture size

In this work, we provide an efficient randomized algorithm to learn the parameters of the distribution. One key component of the algorithm is to project down to a few variables and then learn the parameters from that small projection. For this strategy to work, we must ensure that such restrictions do not collapse the mixture structure—in particular, that the fan-in remains $r$. We show below that such a structural relation always holds in the generic setting.

### 3.1. Small restriction

After preprocessing the input (see Appendix A), we may assume w.l.o.g. that $f$ has no nontrivial common factor and that every variable appears in $f$.

To simplify the parametrization, we apply the shift $x_j \mapsto x_j + 1$ for all $j \in [n]$ in (1). Since we have oracle access to the PGP, we can evaluate the shifted polynomial by querying $f$ at translated inputs. Define $\tilde{f}(x_1, \ldots, x_n) := f(x_1 + 1, \ldots, x_n + 1) = \sum_{i=1}^{r} \mu_i \prod_{j=1}^{n} ((1 - p_{ij})x_j + 1)$. For convenience, we write $t_{ij} := 1 - p_{ij}$, so that each factor becomes $t_{ij}x_j + 1$. Note that after this shift the linear forms are no longer normalized (their coefficients need not sum to 1), but this normalization is not required for our arguments.

Next choose an *alive* set $A \subseteq [n]$ of size $m$, and let $F := [n] \setminus A$ be the set of *fixed* variables. We **emphasize** that the choice of $A$ can be completely deterministic; our guarantees rely on the genericity over the parameters $\{t_{ij}\}$ rather than on the randomness in the restriction. Permuting indices if needed, assume $A = \{1, 2, \ldots, m\}$. We set the fixed variables to 1, i.e., $x_F = \mathbf{1} := (1)_{j \in F}$, and obtain an $m$-variate polynomial $h_A(x_1, \ldots, x_m) := \tilde{f}(x_1, \ldots, x_m, 1, \ldots, 1) = \sum_{i=1}^{r} \mu_i \prod_{j=1}^{m} (t_{ij}x_j + 1)$. From now on we relabel $h_A$ as $h$, so we work with

$$h(x_1, \ldots, x_m) = \sum_{i=1}^{r} \mu_i \prod_{j=1}^{m} (t_{ij}x_j + 1), \qquad (2)$$

where $0 < t_{ij} < 1$ (by Remark 1.1).

The polynomial $h$ is multilinear in $x_1, \ldots, x_m$. Once we have $h$, we compute symbolic rank of the *Jacobian* (see Appendix B for a primer) to argue about the preservation of mixture size.

**Intuition.** Fix an alive set $A$ of size $m$ and view the coefficient vector $C = (c_S)_{S \subseteq [m]}$ of $h$ as a function of the hidden parameters $V := \{\mu_i\}_{i \in [r]} \cup \{t_{ij}\}_{i \in [r], j \in [m]}$. This defines a polynomial map from parameters to observable coefficients. An $r$-mixture has more degrees of freedom than an $(r-1)$-mixture, so generically its image has larger local dimension. We certify this separation by lower-bounding

the symbolic rank of the Jacobian: generically, an $r$-mixture produces a *strictly higher* Jacobian rank than any $(r-1)$-mixture under the same restriction, so the restricted instance still has mixture size $r$.

For any $S \subseteq [m]$, the coefficient of the monomial $x_S := \prod_{j \in S} x_j$ is given by $c_S = \sum_{i=1}^r \mu_i \prod_{j \in S} t_{ij}$. Let $C = (c_S)_{S \subseteq [m]} \in \mathbb{K}^{2^m}$ denote the vector of all coefficients. We only concern ourselves with real distributions and hence $\mathbb{K} = \mathbb{R}$. For each $i \in [r]$ and $S \subseteq [m]$,

$$\frac{\partial c_S}{\partial \mu_i} = \prod_{j \in S} t_{ij}, \qquad \frac{\partial c_S}{\partial t_{i\ell}} = \mu_i \cdot \mathbf{1}_{\{\ell \in S\}} \prod_{j \in S \setminus \{\ell\}} t_{ij}.$$

Viewing the coefficients $\{c_S\}$ as polynomials in the parameter set $V$, we form the Jacobian matrix $J_r$ (as defined in Equation 7) with $2^m$ rows indexed by $S \subseteq [m]$ and $r(m+1)$ columns indexed by variables $v \in V$, with entries $J_{S,v} := \frac{\partial c_S}{\partial v}$.

Let $M_r$ denote the submatrix of $J_r$ obtained by restricting to the columns indexed by $\{t_{i\ell}\}_{i \in [r],\, \ell \in [m]}$ and to the rows indexed by $S \neq \emptyset$. Thus $M_r$ has size $(2^m - 1) \times (mr)$. In $J_r$, the $\mu$-columns contribute at most $r$ additional rank, while the row $S = \emptyset$ contributes one to the rank (because $\partial c_\emptyset / \partial \mu_i = 1$ for all $i$ and $\partial c_\emptyset / \partial t_{i\ell} = 0$). Therefore,

$$1 + \operatorname{rank}(M_r) \leq \operatorname{rank}(J_r) \leq r + \operatorname{rank}(M_r). \quad (3)$$

Hence, it suffices to determine $\operatorname{rank}(M_r)$. In fact, we can show that $\operatorname{rank}(M_r) = rm$ in the generic setting (see Appendix C for the full proof). Thus, we get

$$rm + 1 \leq \operatorname{rank}(J_r) \leq rm + r. \quad (4)$$

**Separating $r$ from $r-1$.** Once we have the above relation, we can separate $J_r$ from $J_{r-1}$ using the rank bounds. For $m \geq r + 1$, the lower bound (4) gives $\operatorname{rank}(J_r) \geq rm + 1$. On the other hand, applying (3) to an $(r-1)$-mixture yields $\operatorname{rank}(J_{r-1}) \leq (r-1) + \operatorname{rank}(M_{r-1}) \leq (r-1) + (r-1)m$, since $M_{r-1}$ has at most $(r-1)m$ columns. Because $m \geq r + 1$, we have $rm + 1 > (r-1)m + (r-1)$, and hence the Jacobian ranks of the $r$- and $(r-1)$-mixture models are separated. This leads us to the following proposition:

**Proposition 3.1.** *For $m \geq r + 1$, in the generic setting, restricting to any alive set of size $m$ does not reduce the top fan-in of the PGC, i.e., the mixture size remains $r$.*

## 4. Moment methods to recover parameters

Anandkumar et al. (2014) introduced the method-of-moments framework for estimating parameters in a broad class of latent-variable models. A key observation is that, for many such models, low-order moments (or cross-moments) admit a *low-rank* tensor structure. For instance, in the single-topic model, these moments form a symmetric tensor and,

in certain regimes, even a symmetric *orthogonally decomposable* tensor (cf. Anandkumar et al. (2014, Theorem 3.1)). Concretely, the second- and third-order moments take the form

$$M_2 = \sum_{i=1}^r \mu_i \, \alpha_i \otimes \alpha_i, \quad (5)$$

$$M_3 = \sum_{i=1}^r \mu_i \, \alpha_i \otimes \alpha_i \otimes \alpha_i, \quad (6)$$

where $\mu_i \in \mathbb{R}$, $\alpha_i \in \mathbb{R}^k$ for all $i \in [r]$, for some $k \in \mathbb{N}$.

To recover the parameters from these moments, one typically assumes a mild non-degeneracy condition: the vectors $\alpha_1, \ldots, \alpha_r$ are linearly independent (and $\mu_i \neq 0$). This implies that $\operatorname{rank}(M_2) = r$, and moreover that $\operatorname{rank}_{\mathrm{CP}}(M_3) = r$, where $\operatorname{rank}_{\mathrm{CP}}(\cdot)$ denotes CP rank (see Kolda & Bader (2009)). When such non-degeneracy conditions fail, learning is conjectured to be computationally hard; see Mossel & Roch (2005); Moitra & Valiant (2010).

*Remark* 4.1. The linear independence of $\alpha_1, \ldots, \alpha_r$ holds *generically*: the set of parameter choices for which the vectors become linearly dependent is characterized by the vanishing of all $r \times r$ minors of the $k \times r$ matrix $(\alpha_1 | \cdots | \alpha_r)$, and hence forms a proper algebraic subset. In particular, this condition holds in our setting as well.

The recovery task can be reduced to a tensor decomposition problem. For $M_2$, (5) corresponds to the eigendecomposition of a rank-$r$ symmetric matrix. While matrix eigendecompositions are well understood, tensor decompositions are not unique in general. However, after *whitening* using $M_2$, the third-order moment becomes an orthogonally decomposable symmetric tensor, and its components can be recovered using the tensor power method (and its robust variants) as in Anandkumar et al. (2014). In our work, we use this tensor-decomposition routine as a black box after extracting $M_2$ and $M_3$ *exactly* from the PGP/PGC representation. We now briefly outline the full procedure.

### 4.1. Extracting moments using PGC

Recall that for any random restriction of size $m$, we can view the restricted distribution as encoded by the restricted PGP $h$ (cf. Equation 2). As discussed above, throughout we assume that the vectors $\alpha_1, \ldots, \alpha_r$ are linearly independent.

In the polynomial $h$, the degree-two terms are exactly the coefficients of monomials $x_i x_j$ for $i \neq j$, and the degree-three terms are exactly the coefficients of monomials $x_i x_j x_k$ with *pairwise distinct* indices. Given a PGP such coefficients can be recovered by a standard interpolation procedure (see Appendix D). Hence, using Lemma D.1, we can recover all such coefficients in time $O(m^3)$. Thus, we obtain all entries $(M_2)_{i,j}$ for $i \neq j$ and all entries $(M_3)_{a,b,c}$ for pairwise distinct $a, b, c \in [m]$.

*Remark* 4.2. We query the PGP *only* for interpolating the coefficients on these small restrictions.

However, to apply the method of moments, we require the full objects $M_2$ and $M_3$, and not just the "off-diagonal" entries. For example, the $(1,1)$ entry of $M_2$ equals: $(M_2)_{1,1} = \sum_{\ell=1}^{r} \mu_\ell \alpha_{\ell,1}^2$. We cannot recover this quantity from interpolation, since the restricted PGP $h$ is multilinear and hence does not contain monomials such as $x_1^2$ (or, more generally, monomials with repeated indices).

In sample-based settings, such missing diagonal (or repeated-index) statistics are typically handled using numerical estimation or optimization procedures. In fact, this issue is well-studied for mixtures of Bernoulli product distributions. Jain & Oh (2014) formulate recovery of the diagonal entries of $M_2$ as a matrix completion problem, and give an alternating minimization algorithm that can exactly recover the diagonal when provided exact off-diagonal entries. For $M_3$, instead of computing it exactly, they estimate a whitened version of $M_3$ via a least-squares method using only the off-diagonal entries. While their algorithm runs in time poly$(n,r)$, the dependence is not linear, and it also requires $M_2$ to be sufficiently well-conditioned.

In contrast, we show that in the "generic" setting, we can recover the missing entries of both $M_2$ and $M_3$ *exactly* and *efficiently* using a purely algebraic procedure. Intuitively, since we have only $r$ mixture components, we have rank$(M_2) \le r$, and similarly every mode-flattening of $M_3$ has rank at most $r$. Therefore, all $(r+1) \times (r+1)$ minors of $M_2$ (and of each flattening of $M_3$) must vanish. By carefully choosing such submatrices so that they contain exactly one unknown entry, each vanishing determinant yields a linear equation in that unknown entry, which we can solve in closed form. If all the parameters had precision upto $\epsilon$, then the determinant of any such matrix will have a precision of at most $O(r^{\frac{r}{2}} 2^r \log \frac{1}{\epsilon})$ (Bareiss, 1972). Hence, we only need to compute it upto $O(r(\log r + \log \frac{1}{\epsilon}))$ bits to compute each of the diagonal entries. This allows us to recover all missing entries of both $M_2$ and $M_3$ in total time $\tilde{O}(r^{3+\omega})$, where $\omega$ is the matrix multiplication exponent abd $\tilde{O}(\cdot)$ is the standard variant of $O(\cdot)$ which subsumes polylogarithmic factors. We defer the technical details to Appendix E, and we summarize the moment extraction procedure in Algorithm 1.

### 4.2. Reconstructing the parameters

We first fix a restriction of size $m$. Then, as described above, we recover $M_2$ and $M_3$ exactly. We now briefly describe how we can learn the parameters using the approach of Anandkumar et al. (2014).

Since $M_2$ is a real symmetric $m \times m$ matrix of rank $r$, it admits an eigendecomposition $M_2 = \sum_{i=1}^{r} d_i \, u_i u_i^\top =$

---

**Algorithm 1** Moment extraction

1: **Input:** PGP $f(x_1, \ldots, x_n)$
2: **Output:** Moment objects $M_2, M_3$
3: Use Lemma D.1 to obtain all off-diagonal entries of $M_2$ and all entries of $M_3$ with pairwise distinct indices.
4: Recover the diagonal entries of $M_2$ (via vanishing $(r+1) \times (r+1)$ minors), as shown in Section E.1.
5: Use vanishing minors of the mode-flattenings of $M_3$ to recover all its remaining entries.

---

$UDU^\top$, where $U \in \mathbb{R}^{m \times r}$ has orthonormal columns, and $D \in \mathbb{R}^{r \times r}$ is diagonal with the (non-zero) eigenvalues $d_1, \ldots, d_r$. We use this eigendecomposition to define a whitening matrix $W \in \mathbb{R}^{m \times r}$ such that $W^\top M_2 W = I_r$. Concretely, define $W = UD^{-1/2}$. Clearly, $W^\top M_2 W = I_r$.

Moreover, using $M_2 = \sum_{i=1}^{r} \mu_i \alpha_i \alpha_i^\top$, we have $W^\top M_2 W = \sum_{i=1}^{r} \mu_i (W^\top \alpha_i)(W^\top \alpha_i)^\top = \sum_{i=1}^{r} \left(\sqrt{\mu_i} W^\top \alpha_i\right)\left(\sqrt{\mu_i} W^\top \alpha_i\right)^\top$. Now set $\hat{\alpha}_i = \sqrt{\mu_i} W^\top \alpha_i \in \mathbb{R}^r$. Then, $I_r = \sum_{i=1}^{r} \hat{\alpha}_i \hat{\alpha}_i^\top$.

Since there are exactly $r$ vectors in $\mathbb{R}^r$ and the above sum has full rank, the vectors $\hat{\alpha}_1, \ldots, \hat{\alpha}_r$ form an orthonormal basis. Worst-case eigen decomposition bounds of $O(m^3)$ for an $m \times m$ matrix are well known (Golub & Van Loan (2013)).

Recall that $M_3 = \sum_{i=1}^{r} \mu_i \alpha_i^{\otimes 3}$, where $M_3$ is an $m \times m \times m$ tensor. We now whiten $M_3$ using $W$ to obtain a smaller $r \times r \times r$ tensor: $\hat{M}_3 := \sum_{i=1}^{r} \mu_i (W^\top \alpha_i)^{\otimes 3} = \sum_{i=1}^{r} \frac{1}{\sqrt{\mu_i}} \hat{\alpha}_i^{\otimes 3}$.

To compute $\hat{M}_3$ from $M_3$, we need to multiply $M_3$ with $W$ along all three modes, i.e., $(\hat{M}_3)_{abc} = \sum_{i=1}^{m} \sum_{j=1}^{m} \sum_{\ell=1}^{m} W_{i,a} W_{j,b} W_{\ell,c} (M_3)_{ij\ell}$. Naively, this computation would take $O(m^3 r^3)$ time. However, we can speed up the whitening using a careful ordering of multiplications (see Appendix F), obtaining $\hat{M}_3$ in time $O(rm^3 + r^2 m^2 + r^3 m)$. Since the precision of all these intermediate computations are determined by the precision $\epsilon$ of our original parameters, they only add log factors in this runtime.

Once we have $\hat{M}_3$, we apply the robust tensor power method of Anandkumar et al. (2014). For any target precision $\epsilon > 0$ and any fixed constant $\delta > 0$, their algorithm runs in time $O(r^{5+\delta}(\log r + \log \log(1/\epsilon)))$ to extract all $r$ eigenvector–eigenvalue pairs, even with repeated eigenvalues.

Finally, using Anandkumar et al. (2014, Theorems 4.1 and 4.3), the robust eigenvectors of $\hat{M}_3$ are precisely the vectors $\hat{\alpha}_i$, with corresponding eigenvalues $\lambda_i = 1/\sqrt{\mu_i}$. Thus, we can recover $\mu_i = 1/\lambda_i^2$. Moreover, since $\hat{\alpha}_i = \sqrt{\mu_i} W^\top \alpha_i$, we can recover $\alpha_i$ via the Moore–Penrose pseudoinverse of $W^\top$: $\alpha_i = (W^\top)^\dagger (1/\sqrt{\mu_i} \hat{\alpha}_i)$.

---

**Algorithm 2** Recovering parameters from moments

1: **Input:** Moment objects $M_2, M_3$.
2: **Output:** Parameters $\{(\mu_i, \alpha_i)\}_{i=1}^r$.
3: Compute the eigendecomposition $M_2 = UDU^\top$ and set $W = UD^{-1/2}$.
4: Whiten $M_3$ to obtain $\hat{M}_3 = M_3(W, W, W)$.
5: Run the tensor power method on $\hat{M}_3$ to obtain robust eigen-pairs $(\lambda_i, \hat{\alpha}_i)$.
6: Set $\mu_i = 1/\lambda_i^2$ and recover $\alpha_i = (W^\top)^\dagger (\hat{\alpha}_i / \sqrt{\mu_i})$.

---

Here we use that each $\alpha_i$ lies in the column span of $M_2$, so the above inversion is well-defined on the relevant subspace. It is known that the complexity of computing the pseudoinverse of an $m \times r$ matrix (with $m \geq r$) is $O(mr^2)$ (Golub & Van Loan (2013)).

Combining all steps, the runtime for these linear-algebraic operations is $O(m^3 + (rm^3 + r^2m^2 + r^3m) + r^{5+\delta}(\log r + \log\log 1/\epsilon) + mr^2)$. Ignoring polylogarithmic factors and for sufficiently small $\delta, \epsilon > 0$, this is $\tilde{O}(rm^3 + r^2m^2 + r^3m + r^{5+\delta})$. Since $m = O(r)$ suffices, the total running time per restriction is $\tilde{O}(r^{5+\delta})$ upto log factors. We describe the procedure in Algorithm 2.

### 4.3. Running time analysis

Using the algorithm of Anandkumar et al. (2014) and our preprocessing step (see Appendix A), we have three main steps: first we recover $M_2, M_3$ fully after extracting the missing entries, we then obtain the whitening matrix $W$ from $M_2$ and use it to produce $\hat{M}_3$, and we then finally run the tensor power method on $\hat{M}_3$ to recover the parameters $\mu_i, \alpha_i$. If we do not use projections, then in settings where $n$ sufficiently dominates $r$, the first step takes $O(n^3)$ time, the second step takes $O(n^3r)$ time, and the final step takes $O(r^{5+\delta}(\log r + \log\log 1/\epsilon))$. In total, we obtain a worst-case running time of $O(n^3r)$, which is practically inefficient when $n$ is large. On the other hand, Jain & Oh (2014) use a matrix completion approach to recover the diagonal entries of the lower-order moments and obtain a runtime that is polynomially dependent on $n, r$, and the condition number $\sigma_1(M_2)/\sigma_r(M_2)$, where $\sigma_i(M_2)$ denotes the $i$-th singular value of $M_2$.

However, if we use our "genericity" conditions along with small projections, then using Proposition 3.1 and the subsequent discussion, we obtain a significantly faster algorithm. As we saw above, we only need $m = O(r)$. Analysing the runtime of all the steps in the oracle access model: $O(n)$ for preprocessing, $\tilde{O}(r^{2+\omega})$ for recovering the moments exactly for each restriction, and then $\tilde{O}(r^{5+\delta})$ for recovering $\mu_i, \alpha_i$ for any restriction. Instead of running the above analysis on $n$-dimensional moments, we run it on restrictions of size $m$ with $m = O(r)$, and hence we need to perform this procedure for only $n/m$ blocks.

---

**Algorithm 3** Main Algorithm

1: **Input:** PGP of $f$
2: **Output:** Parameters $\mu_i, \alpha_{ij}$ for all $i \in [r], j \in [n]$.
3: Use Algorithm 4 to preprocess the input.
4: Set $m = 4r + 1$.
5: **for** $i$ from 1 to $\lfloor \frac{n}{m} \rfloor$ **do**
6:   Pick restriction $S_i$ of size $m$ containing variables $x_i, ..., x_{i+m-1}$ such that $r - 1$ "identifying" variables (using Lemma H.2) are always selected except when $i = 1$. {For $i = \lfloor \frac{n}{m} \rfloor$ pick all the variables until $x_n$.}
7:   Use Algorithm 1 to recover the moments.
8:   Use Algorithm 2 to recover the parameters $\mu_i, \alpha_{ij}$ for $i \in [r], j \in S$.
9: **end for**
10: Stitch together the parameters recovered in each $S_i$ to get complete reconstruction.

---

Combining all this, we obtain a total running time of $\tilde{O}(n/m \cdot (r^{5+\delta} + r^{3+\omega}) + n) = \tilde{O}(n(r^{4+\delta} + r^{2+\omega}))$, for sufficiently small $\delta, \epsilon > 0$. In the general setting, where each query takes time $O(s)$ (where $s$ is the size of the representation), the runtime becomes $\tilde{O}(nsr^2 + n(r^{4+\delta} + r^{2+\omega}))$.

### 4.4. Numerical stability analysis

In this work, we recover the parameters of the distribution exactly, assuming they were specified up to some fixed precision $\epsilon = 2^{-\gamma}$. Indeed, if the parameters were arbitrary irrationals, then no practical algorithm could hope to recover them with infinite precision. Moreover, querying the oracle on an input which marginalizes $n - k$ random variables gives us an output with $2^{-k\gamma + \log r}$ precision. Not that in our algorithm, we never query the oracle on an input with less than $n - 3$ marginalized random variables. Hence, at any point we only need to store the oracle output up to $O(\log r + \log \frac{1}{\epsilon})$ bits. Further, to ensure that the genericity condition holds with high probability, we need the number of bits $\gamma = \log \frac{1}{\epsilon} > O(r \log n)$. This is because the total degree of all the algebraic conditions on the parameters (dominated by the linear independence of the $\{\alpha_i\}_{i=1}^r$) is $O(rn^r)$ and then by Lemma A.1, we need the parameters to come from a set of size atleast twice this size and only need $O(r \log n)$ bits.

The main reconstruction step in our algorithm uses the tensor power method as used in Anandkumar et al. (2014). In their work, they show that the method has a quadratic rate of convergence if we start with a "good" vector. Formally, if $e_t$ denotes the error at iteration $t$, then one has a recurrence of the form $\|e_{t+1}\| \leq C\|e_t\|^2$ for some constant $0 < C < 1$. In particular, the error decays doubly-exponentially with $t$. Hence, we get the $\log\log \frac{1}{\epsilon}$ factor in the runtime. However, we still need to get this "good" starting vector $\theta$. Anandkumar et al. (2014) show that with probability $\geq 1 - \eta$, we can get such a $\theta$ in $r^{1+\delta} \log 1/\eta$ iterations. They also show that in $\log r$ iterations, starting from $\theta$ we get $\theta'$ which is a

vector that is in the zone of quadratic convergence. Combining this with the $O(r^3)$ running time for each iteration for each of the $r$ eigenvectors, we get a randomized algorithm with total running time of $O(r^{5+\delta}(\log r + \log \log \frac{1}{\epsilon}))$ which recovers all parameters exactly if they were specified up to $\epsilon$ precision.

Finally, we will also like to make the following remark. Gordon et al. (2021) show that source identification is possible if and only if at least $2r - 1$ random variables (RVs) have mixture probabilities that are mutually distinct and separated by $\epsilon$. In this case, they give a $n^{O(r)}2^{r^2}$ query complexity algorithm for identification. Further, if they have all RVs $\epsilon$-separated then they can give a $O(n2^{r^2})$ query complexity algorithm. In the same context, if we have all random variables as $\epsilon$-separated our algorithm will return the parameters up to $\epsilon$ precision in $O(nr^2)$ query complexity. When instead only $2r - 1$ random variables are $\epsilon$-separated, our method performs substantially better than Gordon et al. (2021). An implication of Lemma H.2 is that if one fixed variable, say $X_1$, is known to be $\epsilon$-separated, then keeping $X_1$ alive in every restriction already suffices to distinguish all mixture components since $\alpha_{i,1} \neq \alpha_{j,1}$ for all $i \neq j \in [r]$. The main challenge is that we do not know beforehand which variables are the $\epsilon$-separated ones. In Gordon et al. (2021), this necessitates a brute-force search over all candidate subsets of size $2r - 1$, giving rise to the $n^{O(r)}$ overhead. By contrast, our algorithm needs only *one* good variable! So, it suffices to brute-force over the $n$ possible choices of that variable. As a result, when only $2r - 1$ variables are $\epsilon$-separated, we obtain query complexity $O(n^2r^2)$ where each parameter is recovered up to $\epsilon$ precision, significantly improving over their work.

## 5. Conclusion and future directions

In this work, to the best of our knowledge, we give the first algorithm that requires *linear* in $n$ and *quadratic* in $r$ queries to exactly learn the parameters defining a mixture of products of Bernoulli distributions.

We believe that there are many interesting future directions to consider. One of them is extending this setup to the noisy setting. Concretely, we can assume some bounded noise in the PGC outputs and then see how efficiently and closely we can reconstruct the parameters. Secondly, we want to explore reconstruction when, on a certain fraction of inputs, the PGP gives adversarial oracle outputs while being correct on the remaining queries. Finally, reconstruction for different types of distributions like mixtures of bounded-treewidth models, and sparse distributions represented by PGPs is also an interesting open question.

## Acknowledgements

Part of this work was supported by the Simons Institute for the Theory of Computing, and conducted when MB and PD were visiting the Institute as participants of the program "Complexity and Linear Algebra". We would like to thank the anonymous reviewers for their helpful feedback. PD is supported by the SUG Grant (#025774-00001), funded by Nanyang Technological University.

## Impact Statement

This paper presents work whose goal is to advance the field of Machine Learning. There are many potential societal consequences of our work, none which we feel must be specifically highlighted here.

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

# A. Pre-processing: Identifying common factors

Since we have oracle access to the probability generating polynomial (PGP), we will rely on standard tools from algebraic complexity for our algorithm. In particular, we use Polynomial Identity Testing (PIT).

## A.1. Polynomial Identity Testing (PIT)

The PIT problem asks whether a given polynomial is identically zero. Explicitly expanding a multivariate polynomial may require inspecting exponentially many monomials, but PIT can be performed efficiently via randomized evaluation: a nonzero polynomial remains nonzero under a random substitution with high probability.

**Lemma A.1** (Zippel 1979, Schwartz 1980). *Let $p(x_1, \ldots, x_n)$ be a nonzero polynomial of total degree at most $d$ over a field $\mathbb{K}$. Let $S \subseteq \mathbb{K}$ be a finite set, and sample $a_1, \ldots, a_n \in S$ uniformly at random. Then*

$$\Pr\big[p(a_1, \ldots, a_n) \neq 0\big] \;\geq\; 1 - \frac{d}{|S|}.$$

## A.2. Preprocessing step

Given $f$ as in Equation 1, in this step, we reduce the effective number of variables by (i) removing variables that do not appear in $f$, and (ii) factoring out any linear term common to all mixture components.

**Variables not appearing in $f$.** A variable $x_j$ does not appear in $f$ if and only if $f(1, \ldots, 1) = f(1, \ldots, 1, 0, 1, \ldots, 1)$, where the 0 occurs in the $j$-th position. This is because, using Equation 1 and Remark 1.1, $f(1, \ldots, 1, 0, 1, \ldots, 1) = \sum_i \mu_i p_{ij} < \sum_i \mu_i = f(1, \ldots, 1)$. Let $S_{\text{not}} \subseteq [n]$ be the set of such indices. This can be found using $1 + n$ queries: first query $f(1, \ldots, 1)$ once, and then query $f$ with $x_j = 0$ and all other variables set to 1 for each $j$.

**Common linear factors.** Fix $j \notin S_{\text{not}}$ and define the univariate polynomial

$$g_j(x) := f(1, \ldots, 1, x, 1, \ldots, 1).$$

By Equation 1, $g_j$ is affine in $x$ and $g_j(1) = f(1, \ldots, 1) \neq 0$. Set

$$p_j := \frac{g_j(0)}{g_j(1)} \qquad \text{and} \qquad z_j := -\frac{p_j}{1 - p_j}.$$

Then $(1 - p_j)x_j + p_j$ divides $f$ if and only if the polynomial obtained by substituting $x_j = z_j$ is identically zero:

$$f(x_1, \ldots, x_{j-1}, z_j, x_{j+1}, \ldots, x_n) \equiv 0.$$

We can test this condition using PIT (Lemma A.1). Let $S_f \subseteq [n]$ denote the set of indices for which this test succeeds, and store the corresponding parameters $p_j$ for each $j \in S_f$.

Overall, this preprocessing uses $O(n)$ oracle evaluations and identifies all such common linear factors along with their exact parameters. Henceforth, whenever we query the PGP, we will set all variables in

$$S_b := S_f \cup S_{\text{not}}$$

to 1 and work with the resulting reduced polynomial on $[n] \setminus S_b$.

The details have been summarized in Algorithm 4. In the PIT check in the algorithm, we view $f(x_1, \ldots, x_{j-1}, z_j, x_{j+1}, \ldots, x_n)$ as a polynomial in the remaining variables and test whether it is identically zero using Lemma A.1, which is just evaluation of the poynomial at some random points, which is $O(1)$ in the oracle access model, and $O(s)$ in the general setting, where $s$ is the size of the PGC.

It is easy to see that we only make $O(n)$ queries to the PGP in the algorithm. Further, in each iteration, we do at most one PIT check. Hence, the query complexity is $O(n)$. The running time is $O(n)$ in oracle access model, and $O(ns)$ in the general case.

---

**Algorithm 4** Preprocessing: removing irrelevant variables and identifying all common factors

---

1: **Input:** Oracle access to $f(x_1, \ldots, x_n)$
2: **Output:** $S_{\text{not}} \subseteq [n]$, $S_f \subseteq [n]$, and parameters $\{p_j\}_{j \in S_f}$
3: Query $F \leftarrow f(1, 1, \ldots, 1)$
4: $S_{\text{not}} \leftarrow \emptyset, \quad S_f \leftarrow \emptyset$
5: **for** $j = 1$ to $n$ **do**
6:      Query $F_j \leftarrow f(1, \ldots, 1, \underbrace{0}_{j\text{-th}}, 1, \ldots, 1)$
7:      **if** $F_j = F$ **then**
8:          $S_{\text{not}} \leftarrow S_{\text{not}} \cup \{j\}$ $\{x_j$ does not appear in $f\}$
9:      **else**
10:         $p_j \leftarrow F_j / F$
11:         $z_j \leftarrow -p_j / (1 - p_j)$
12:         **if** $f(x_1, \ldots, x_{j-1}, z_j, x_{j+1}, \ldots, x_n) \equiv 0$ **(via Lemma A.1) then**
13:            $S_f \leftarrow S_f \cup \{j\}$
14:            Store parameter $p_j$
15:         **end if**
16:      **end if**
17: **end for**

---

## B. Jacobian and symbolic rank

Let $\Phi = (\Phi_1, \ldots, \Phi_k) : \mathbb{K}^d \to \mathbb{K}^k$ be a polynomial map over a field $\mathbb{K}$, and let $\theta = (\theta_1, \ldots, \theta_d) \in \mathbb{K}^d$ denote the parameter vector (in our setting, $\theta$ collects parameters such as $\mu_i$ and $t_{ij}$). The *Jacobian* of $\Phi$ is the $k \times d$ matrix

$$J_\Phi(\theta) := \left( \frac{\partial \Phi_a}{\partial \theta_b} \right)_{a \in [k], \, b \in [d]}, \tag{7}$$

which we view as a *symbolic* matrix with entries in the polynomial ring $\mathbb{K}[\theta]$. Its *symbolic rank* is the rank of $J_\Phi(\theta)$ over the function field $\mathbb{K}(\theta)$, equivalently the largest $t$ such that some $t \times t$ minor is not the zero polynomial. In particular, if $J_\Phi(\theta)$ has full column rank symbolically, then it has full column rank for all parameter values outside a proper Zariski-closed set, yielding a generic non-degeneracy condition (Vakil, 2025; Leung et al., 2016). When $\mathbb{K} \in \{\mathbb{R}, \mathbb{C}\}$, full column rank at a point further implies local injectivity of $\Phi$ in a neighborhood of that point by the inverse/implicit function theorem (Lee, 2012).

*Remark* B.1. In our setting, we first apply a restriction to project the PGP to $m = O(r)$ variables, and then view the resulting low-degree coefficients (e.g., all degree-$\leq 3$ coefficients) as polynomial functions of the underlying parameters $\theta$. This defines a polynomial map $\Phi$ from parameters to observable coefficients. We analyze the symbolic Jacobian of this map to show that it has *large* rank generically, which in turn implies that the restricted instance remains identifiable for almost all parameter choices.

## C. Rank bound on the Jacobian

In this section we show how we can bound the rank of the Jacobian matrix $J_r$ formed from the distribution coefficients of a $r$-mixture. We demonstrate that there is a separation in these ranks for any $r_1 \neq r_2$. For ease of reading, we begin with the special case $r = 2$, which serves as a gentle introduction to the main ideas.

### C.1. Toy case: separating $r = 2$ from $r = 1$

In this subsection we illustrate the main idea behind our rank-based separation argument by analyzing the smallest nontrivial case: distinguishing fan-in 2 from fan-in 1. The general rank bound for arbitrary $r$ is proved later.

W.l.o.g. let $A = [m]$. For $r = 1$, the parameters are $\mu, t_1, \ldots, t_m$ (for simplicity), and the Jacobian $J_1$ (with rows indexed by $S \subseteq [m]$ and columns indexed by $\{\mu\} \cup \{t_j\}_{j \in [m]}$) has $2^m$ rows and $m + 1$ columns. In particular, rank$(J_1) \leq m + 1$, and in fact, it is easy to show that rank$(J_1) = m + 1$. Consider the $(m + 1) \times (m + 1)$ submatrix of $J_1$ obtained by taking rows indexed by $S = \emptyset, \{1\}, \ldots, \{m\}$ and columns indexed by $\mu, t_1, \ldots, t_m$. This submatrix is upper triangular with diagonal entries $1, \mu, \ldots, \mu$, and hence has determinant $\mu^m \neq 0$ as a polynomial.

*Fact* 1. The rank of $J_1$ is $m + 1$.

For $r = 2$, the Jacobian $J_2$ has $2^m$ rows and $2(m+1)$ columns. It has the following form:

$$
\begin{array}{c}
\\
c_\emptyset \\
c_{\{1\}} \\
\vdots \\
c_{\{1,2\}} \\
\vdots \\
c_{[m]}
\end{array}
\begin{array}{c}
\mu_1 \quad \mu_2 \quad t_{11},\ldots,t_{1m} \quad t_{21},\ldots,t_{2m} \\
\left(
\begin{array}{cccc}
1 & 1 & 0,\ldots,0 & 0,\ldots,0 \\
* & * & v_{1,\{1\}} & v_{2,\{1\}} \\
\vdots & \vdots & \vdots & \vdots \\
* & * & v_{1,\{1,2\}} & v_{2,\{1,2\}} \\
\vdots & \vdots & \vdots & \vdots \\
* & * & v_{1,[m]} & v_{2,[m]}
\end{array}
\right)
\end{array}
\tag{8}
$$

where $v_{k,S}$ denotes the row vector corresponding to derivatives of $c_S$ with respect to the variables $\{t_{k1},\ldots,t_{km}\}$. For example, for $S = \{1, 2\}$ we have $c_S = \mu_1 t_{11} t_{12} + \mu_2 t_{21} t_{22}$, and hence $v_{1,S} = [\mu_1 t_{12} \ \ \mu_1 t_{11} \ \ 0 \ \cdots \ 0]$.

We now isolate the contribution of the constant coefficient row $c_\emptyset$. Using row operations, we can make all entries in the first $\mu_1$-column zero except in the first row. Next, using column operations among the $\mu_1, \mu_2$ columns, we can make the top entry of the $\mu_2$-column equal to 0. Since all $t$-columns have a 0 in the first row, these operations do not affect any $v_{k,S}$ block. As a result, the rank of $J_2$ is at least *one more* than the rank of the submatrix obtained by restricting to: (i) rows indexed by nonempty sets $S \neq \emptyset$, and (ii) columns indexed only by the $t_{ij}$'s.

**Proposition C.1.** *Let $M$ be the submatrix of $J_2$ whose rows are indexed by nonempty $S \subseteq [m]$ and whose columns are indexed by $\{t_{ij}\}_{i\in[2],\,j\in[m]}$. Then*

$$
rank(J_2) \ \geq \ 1 + rank(M).
$$

We now lower bound $\text{rank}(M)$. Group the rows of $M$ according to $|S|$, and view $M$ in the following block form:

$$
M = \begin{array}{c}
|S|=1 \\
|S|=2 \\
\vdots \\
|S|=m
\end{array}
\begin{array}{c}
(t_{1j})_{j\in[m]} \quad (t_{2j})_{j\in[m]} \\
\left(
\begin{array}{cc}
\mu_1 I_m & \mu_2 I_m \\
\mu_1 B_{1,2} & \mu_2 B_{2,2} \\
\vdots & \vdots \\
\mu_1 B_{1,m} & \mu_2 B_{2,m}
\end{array}
\right)
\end{array}
$$

where $B_{i,k}$ is a $\binom{m}{k} \times m$ matrix whose entries are polynomials in $\{t_{i,j}\}_{j\in[m]}$. Concretely, the rows of $B_{i,k}$ are the vectors $\frac{1}{\mu_i} v_{i,S}$ over all $k$-subsets $S \subseteq [m]$.

Since we only care about rank (and work over a field), we may apply row and column operations; in particular, we may assume $\mu_1 \neq 0$ (otherwise swap the two mixture components). Perform the following column operations: for each $j \in [m]$, replace the column $t_{2j}$ by $t_{2j} \leftarrow t_{2j} - \frac{\mu_2}{\mu_1} t_{1j}$. This makes the singleton block in the $(t_{2j})$ columns identically zero, while the $\mu_1 I_m$ block in the $(t_{1j})$ columns remains unchanged. Hence $\text{rank}(M) \geq m$.

Next consider any doubleton row $S = \{a, b\}$. Before the column operation, the entry in this row and the column $t_{2a}$ equals $\mu_2 t_{2b}$, while the entry in the same row and the column $t_{1a}$ equals $\mu_1 t_{1b}$. After the column operation, the new entry in column $t_{2a}$ becomes $\mu_2 t_{2b} - \frac{\mu_2}{\mu_1} \cdot (\mu_1 t_{1b}) = \mu_2(t_{2b} - t_{1b})$, which is a nonzero polynomial. Therefore, the transformed $(t_{2j})$ block contains a nonzero entry in some doubleton row, while all singleton rows have zeros in the $(t_{2j})$ block. This yields at least one additional independent direction beyond the $m$ pivots coming from the $\mu_1 I_m$ block, implying $\text{rank}(M) \geq m + 1$. Combining with Proposition C.1, we conclude that $\text{rank}(J_2) \geq m + 2$, separating the $r = 2$ case from the $r = 1$ case.

*Fact 2.* The rank of $J_2$ is at least $m + 2$.

### C.2. Rank bound on the Jacobian in the general setting

We now prove rank-separation in the general $r$ case. Recall that the Jacobian $J_r$ has rows indexed by $S \subseteq [m]$ and columns indexed by the parameters $\{\mu_i\}_{i\in[r]} \cup \{t_{i\ell}\}_{i\in[r],\,\ell\in[m]}$. Let $M$ denote the submatrix of $J_r$ obtained by columns indexed by $\{t_{i\ell}\}_{i\in[r],\,\ell\in[m]}$, rows indexed by $S$ (except $S = \emptyset$). Thus $M$ has size $2^m - 1 \times mr$ and entries

$$
M_{S,(i,\ell)} \ = \ \frac{\partial c_S}{\partial t_{i\ell}} \ = \ \mu_i \cdot \mathbf{1}_{\{\ell \in S\}} \prod_{j \in S\setminus\{\ell\}} t_{ij}.
$$

In $J_r$, the $\mu$-columns contribute at most $r$ additional rank, while the row $S = \emptyset$ contributes at least one rank (because $\partial c_\emptyset / \partial \mu_i = 1$ for all $i$ and $\partial c_\emptyset / \partial t_{i\ell} = 0$). Therefore,

$$1 + \text{rank}(M) \ \leq \ \text{rank}(J_r) \ \leq \ r + \text{rank}(M). \tag{9}$$

Hence, it suffices to determine $\text{rank}(M)$.

**A specialization.**   To lower bound the *symbolic* rank of $M$, it is enough to exhibit one assignment of parameters for which $M$ attains large rank. We specialize to

$$t_{ij} = a_i \cdot b_j \qquad \text{for all } i \in [r], \ j \in [m], \tag{10}$$

where $a_i, b_j \neq 0$. (Since we only use this as a witness assignment for rank, the signs are irrelevant; in our model one may take $a_i, b_j > 0$.)

Under (10), for any $S \subseteq [m]$ with $|S| = k$ and any $\ell \in [m]$,

$$\frac{\partial c_S}{\partial t_{i\ell}} \ = \ \mu_i \cdot \mathbf{1}_{\{\ell \in S\}} \prod_{j \in S \setminus \{\ell\}} (a_i b_j) \ = \ \mu_i \, a_i^{k-1} \cdot \mathbf{1}_{\{\ell \in S\}} \prod_{j \in S \setminus \{\ell\}} b_j.$$

Grouping the rows of $M$ by $k = |S|$, we can write $M$ in the following block form:

$$M = \begin{array}{c} \\ |S| = 1 \\ |S| = 2 \\ \vdots \\ |S| = k \\ \vdots \\ |S| = m \end{array} \begin{pmatrix} (t_{11}, \ldots, t_{1m}) & (t_{21}, \ldots, t_{2m}) & \cdots & (t_{r1}, \ldots, t_{rm}) \\ \mu_1 I_m & \mu_2 I_m & \cdots & \mu_r I_m \\ \mu_1 a_1 G_2 & \mu_2 a_2 G_2 & \cdots & \mu_r a_r G_2 \\ \vdots & \vdots & \ddots & \vdots \\ \mu_1 a_1^{k-1} G_k & \mu_2 a_2^{k-1} G_k & \cdots & \mu_r a_r^{k-1} G_k \\ \vdots & \vdots & \ddots & \vdots \\ \mu_1 a_1^{m-1} G_m & \mu_2 a_2^{m-1} G_m & \cdots & \mu_r a_r^{m-1} G_m \end{pmatrix} \tag{11}$$

where for each $k \in \{1, \ldots, m\}$ the matrix $G_k$ is a $\binom{m}{k} \times m$ matrix depending only on $(b_1, \ldots, b_m)$, with rows indexed by $k$-subsets $S$ and columns indexed by $\ell \in [m]$, and entry

$$(G_k)_{S,\ell} \ = \ \mathbf{1}_{\{\ell \in S\}} \prod_{j \in S \setminus \{\ell\}} b_j.$$

Note that $G_1 = I_m$, and $G_m$ is a $1 \times m$ row vector with all entries nonzero.

**Lemma C.2.** *For every $k \in \{1, 2, \ldots, m-1\}$, we have $\text{rank}(G_k) = m$.*

*Proof.* We prove the claim by exhibiting one assignment of $(b_1, \ldots, b_m)$ for which $G_k$ has full column rank. Set $b_1 = \cdots = b_m = 1$. Then $(G_k)_{S,\ell} = \mathbf{1}_{\{\ell \in S\}}$, i.e., $G_k$ becomes the incidence matrix of $k$-subsets vs. elements.

Suppose $\alpha \in \mathbb{K}^m$ satisfies $G_k \alpha = 0$. Then for every $k$-subset $S \subseteq [m]$ we have $\sum_{\ell \in S} \alpha_\ell = 0$. Fix $i \neq j$ and choose a set $T \subseteq [m] \setminus \{i, j\}$ of size $k - 1$ (possible since $k \leq m - 1$). Apply the constraint to $S = T \cup \{i\}$ and $S' = T \cup \{j\}$ and subtract to obtain $\alpha_i - \alpha_j = 0$. Thus all coordinates of $\alpha$ are equal: $\alpha_1 = \cdots = \alpha_m = \alpha$. Plugging into any constraint gives $k\alpha = 0$, hence $\alpha = 0$ (over characteristic 0, e.g. $\mathbb{R}$ or $\mathbb{C}$). Therefore the nullspace is trivial and $\text{rank}(G_k) = m$. $\square$

**Reducing to a Vandermonde structure.**   By Lemma C.2, for each $k \in \{1, \ldots, m-1\}$ there exists an invertible matrix of row operations $R_k$ such that $R_k G_k$ has the form $\begin{bmatrix} I_m \\ 0 \end{bmatrix}$ (i.e., its first $m$ rows form $I_m$). Applying these row operations independently within each $|S| = k$ block of (11) (and discarding the resulting all-zero rows), we obtain a submatrix $M'$ of

$M$ with $(m-1) \cdot m + 1$ rows and $rm$ columns of the form

$$
M' = 
\begin{array}{c}
\\
k=1 \\
k=2 \\
\vdots \\
k=m-1 \\
k=m
\end{array}
\begin{array}{c}
(t_{11},\ldots,t_{1m}) \quad (t_{21},\ldots,t_{2m}) \quad \cdots \quad (t_{r1},\ldots,t_{rm}) \\
\left(
\begin{array}{cccc}
\mu_1 I_m & \mu_2 I_m & \cdots & \mu_r I_m \\
\mu_1 a_1 I_m & \mu_2 a_2 I_m & \cdots & \mu_r a_r I_m \\
\vdots & \vdots & \ddots & \vdots \\
\mu_1 a_1^{m-2} I_m & \mu_2 a_2^{m-2} I_m & \cdots & \mu_r a_r^{m-2} I_m \\
\mu_1 a_1^{m-1}\tilde{u} & \mu_2 a_2^{m-1}\tilde{u} & \cdots & \mu_r a_r^{m-1}\tilde{u}
\end{array}
\right)
\end{array}
\tag{12}
$$

where $\tilde{u}$ is a fixed nonzero $m \times 1$ vector (coming from $G_m$). Clearly $\mathrm{rank}(M) = \mathrm{rank}(M')$ and hence $\mathrm{rank}(M) \leq \min\{rm, (m-1)m+1\}$.

Now assume $m \geq r+1$ and choose $\mu_i \neq 0$ and pairwise distinct $a_1, \ldots, a_r$. Consider the submatrix of $M'$ formed by taking the first $r$ block-rows (i.e., $k = 1, 2, \ldots, r$) and all columns. This submatrix equals

$$
\left(
\begin{bmatrix}
1 & 1 & \cdots & 1 \\
a_1 & a_2 & \cdots & a_r \\
\vdots & \vdots & \ddots & \vdots \\
a_1^{r-1} & a_2^{r-1} & \cdots & a_r^{r-1}
\end{bmatrix}
\otimes I_m
\right)
\cdot \mathrm{diag}(\mu_1, \ldots, \mu_r),
$$

which has rank $rm$ since the $r \times r$ Vandermonde matrix is nonsingular when the $a_i$'s are distinct. Therefore $\mathrm{rank}(M') = rm$, and hence $\mathrm{rank}(M) = rm$ for this assignment. Consequently, the symbolic rank of $M$ is $rm$ whenever $m \geq r+1$.

Combining with (3), for $m \geq r+1$ we obtain

$$
rm + 1 \;\leq\; \mathrm{rank}(J_r) \;\leq\; rm + r. \tag{13}
$$

*Remark* C.3. The lower bound above fails only on a proper algebraic subset of the parameter space (where all $rm \times rm$ minors of $M$ vanish). Hence, in the generic setting and for $m \geq r+1$, the ranks of the Jacobians separate different mixture sizes. In particular, this separation holds for arbitrary choices of the weights $\{\mu_i\}$ (as long as $\mu_i \neq 0$), since $\mu$ only appears as a nonzero diagonal scaling in the Vandermonde witness argument above.

## D. Interpolation

Given query access to a polynomial $f$, a basic task is to recover its low-degree coefficients. By Equation 1, our PGP is multilinear (i.e., each variable appears with exponent either $0$ or $1$ in any monomial). In our algorithm we will need to extract all degree-2 and degree-3 coefficients.

**Lemma D.1.** *Suppose we have query access to a multilinear polynomial $f \in \mathbb{C}[x_1, \ldots, x_n]$ of degree at most $n$:*

$$
f(x_1, \ldots, x_n) = \sum_{S \subseteq [n]} c_S \prod_{i \in S} x_i.
$$

*Then we can recover $c_S$ for all $|S| \leq 3$ using $O(n^3)$ queries.*

*Proof.* For each $S \subseteq [n]$, let $e_S \in \{0,1\}^n$ denote the indicator vector of $S$, i.e., $e_S(j) = 1$ if $j \in S$ and $e_S(j) = 0$ otherwise. Since $f$ is multilinear,

$$
f(e_S) = \sum_{T \subseteq S} c_T.
$$

In particular, $c_\emptyset = f(e_\emptyset) = f(0, \ldots, 0)$. For each $i \in [n]$,

$$
c_{\{i\}} = f(e_{\{i\}}) - c_\emptyset.
$$

For each distinct $i, j \in [n]$,

$$
c_{\{i,j\}} = f(e_{\{i,j\}}) - c_{\{i\}} - c_{\{j\}} + c_\emptyset.
$$

For each distinct $i, j, k \in [n]$,

$$c_{\{i,j,k\}} = f(e_{\{i,j,k\}}) - c_{\{i,j\}} - c_{\{i,k\}} - c_{\{j,k\}} - c_{\{i\}} - c_{\{j\}} - c_{\{k\}} - c_\emptyset.$$

Thus, by querying $f(e_S)$ for all $S$ of size at most 3 and storing these values, we can compute all coefficients $c_S$ with $|S| \leq 3$. The number of such subsets is $1 + \binom{n}{1} + \binom{n}{2} + \binom{n}{3} = O(n^3)$. □

*Remark* D.2. We will only apply Lemma D.1 after restricting $f$ to $m = O(r)$ variables, so the cost becomes $O(m^3) = \text{poly}(r)$ per restriction.

## E. Recovering missing entries of $M_2$ and $M_3$ from off-diagonal entries

We show here how, in the generic setting, we can recover the missing diagonal (and repeated-index) entries of $M_2$ and $M_3$ using only the off-diagonal entries obtained via interpolation. Formally, we show the following.

**Proposition E.1.** *Let $M$ be a real symmetric matrix of size $m \times m$ and rank $r$, and $T$ be a real third-order symmetric tensor of size $m \times m \times m$ and CP rank $r$.*

1. *Suppose we have access to all off-diagonal entries of $M$, ie, we are given $M_{i,j}$ for all $i \neq j$ where $i, j \in [m]$. Then, in the generic setting, we can recover all the diagonal entries of $M$ ($M_{i,i}$ for $i \in [m]$) in time $\tilde{O}(m^2 + mr^{1+\omega})$ where $\omega$ is the matrix multiplication exponent.*

2. *Suppose we have access to all off-diagonal entries of $T$, ie, we are given $T_{i,j,k}$ for all pairwise distinct $i, j, k \in [m]$. Then, in the generic setting, we can recover all the diagonal and repeated index entries of $T$ ($T_{i,j,k}$ for $i, j, k \in [m]$ where at least two of the three indices are equal) in time in time $\tilde{O}(m^3 + m^2 r^{1+\omega})$.*

*Further, these entries are unique in the generic setting. Here, $\tilde{O}(\cdot)$ denotes standard complexity variant of $O(\cdot)$ which subsumes polylogarithmic factors.*

The proof of the proposition follows in the subsequent subsections.

### E.1. Vanishing of minors of a symmetric rank-$r$ matrix in the generic setting

We first make the following observation: Let $A$ be a real symmetric $n \times n$ matrix of rank exactly $r$. Then, *generically* over the set of rank-$r$ symmetric matrices, all $r \times r$ minors of $A$ are non-zero, otherwise they would form a proper algebraic subset over such matrices. In particular, consider a Vandermonde-type matrix $V \in \mathbb{R}^{n \times r}$ defined by choosing distinct nonzero scalars $t_1, \ldots, t_n$ and setting $V_{p,q} = t_p^{q-1}$. For any $I \subseteq [n]$ with $|I| = r$, the submatrix $V_I$ (rows indexed by $I$) is a Vandermonde matrix on distinct parameters and hence satisfies $\det(V_I) \neq 0$. Further, for $M = VV^T$, since $V^T$ has full-column rank $r$, $\text{rank}(M) = r$. Hence, it is a *proper* subset.

This is useful for reconstructing the missing diagonal entries of $M_2$ from its off-diagonal entries. Fix a restriction of size $m \geq 2r + 1$ and obtain all off-diagonal entries of $M_2$ (e.g., by interpolation on degree-2 coefficients). Now choose two disjoint $r$-subsets $A_1, A_2 \subseteq [m]$ such that $A_1 \cap A_2 = \emptyset$, and pick an index

$$\ell \in [m] \setminus (A_1 \cup A_2).$$

Consider the $(r + 1) \times (r + 1)$ submatrix $K$ of $M_2$ whose rows are indexed by $A_1 \cup \{\ell\}$ and whose columns are indexed by $A_2 \cup \{\ell\}$. By construction, $K$ contains exactly one diagonal entry of $M_2$, namely

$$x := (M_2)_{\ell,\ell},$$

and all other entries of $K$ are off-diagonal and hence already known.

Since $\text{rank}(M_2) = r$, every $(r + 1) \times (r + 1)$ minor vanishes, so $\det(K) = 0$. Expanding $\det(K)$ along the row/column containing $x$, we obtain an identity of the form

$$x C_1 + C_2 = 0,$$

where $C_1$ is an $r \times r$ minor of $M_2$ (specifically, the cofactor corresponding to $x$), and $C_2$ depends only on the known off-diagonal entries. By the observation above, generically $C_1 \neq 0$, and hence we can solve uniquely for

$$x = -\frac{C_2}{C_1}.$$

To compute $x$ numerically, it suffices to compute $C_1$ (a determinant of size $r$) and $C_2$ (which can be obtained, for example, by evaluating $\det(K)$ after setting $x = 0$, i.e., one determinant of size $r + 1$). Thus, the time to recover a single diagonal entry is $T_r + T_{r+1}$, where $T_k$ denotes the time to compute the determinant of a $k \times k$ numerical matrix. Using fast linear algebra, $T_k = O(k^\omega)$. Further, if all the parameters had precision upto $\epsilon$, then each off-diagonal entry of of $M_2$ can have precision upto $\epsilon^2$. The determinant of any such matrix will have a precision of at most $O(r^{\frac{r}{2}} 2^r \log \frac{1}{\epsilon}^2)$ (Bareiss, 1972). Hence, we need to compute it upto $O(r(\log r + \log \frac{1}{\epsilon}))$ bits to compute the diagonal entries. Hence, recovering all $m$ diagonal entries takes total time $O(mr^{1+\omega}(\log r + \log \frac{1}{\epsilon}))$.

Using Lemma D.1, we only need $O(m^2)$ queries to obtain all off-diagonal entries of $M_2$. In case of oracle access to the PGP this leads to time complexity of $O(m^2)$, while in the general case it is $O(sm^2)$, where $s$ is the size of the PGC. Combining this with the time to reconstruct the diagonal entries, the total time to recover the full *exact* matrix $M_2$ is

$$\tilde{O}(m^2 + mr^{1+\omega})$$

in the oracle access model, and $\tilde{O}(sm^2 + mr^{1+\omega})$ in the general case. Further, the query complexity is $O(m^2)$ for a fixed alive set.

### E.2. Vanishing of minors of a third-order rank $r$ tensor in the generic setting

In $M_2$, interpolation helps us recover all off-diagonal entries. For $M_3$ as above, we can again recover all entries $(M_3)_{a,b,c}$ for all *pairwise distinct* $a, b, c$ by interpolation. However, we are again missing entries where $a = b = c$, or where exactly two of the three indices coincide. We show that using similar ideas as above, we can recover these entries *exactly*.

Let $T \in \mathbb{R}^{n \times n \times n}$ be a third-order symmetric tensor of the form

$$T = \sum_{i=1}^{r} a_i \, u_i \otimes u_i \otimes u_i, \qquad a_i \neq 0, \tag{14}$$

where $u_1, \ldots, u_r \in \mathbb{R}^n$ are linearly independent. Let $U = [u_1 \cdots u_r] \in \mathbb{R}^{n \times r}$ and $\Lambda = \mathrm{diag}(a_1, \ldots, a_r)$. Using the matricized CP identity (Kolda & Bader, 2009, Equation (3.2)), the mode-1 unfolding of $T$ can be written as

$$T_{(1)} = U \Lambda (U \odot U)^\top,$$

where $\odot$ denotes the Khatri–Rao product (columnwise Kronecker product). Note that $(U \odot U) \in \mathbb{R}^{n^2 \times r}$ is precisely the matrix

$$L = [\, u_1 \otimes u_1 \ \ u_2 \otimes u_2 \ \ \cdots \ \ u_r \otimes u_r \,].$$

In particular, $L$ has at most $r$ columns, and in the generic setting these columns are linearly independent, i.e., $\mathrm{rank}(L) = r$.[2] Since $\mathrm{rank}(U) = r$ and $\mathrm{rank}(\Lambda) = r$, we obtain

$$\mathrm{rank}(T_{(1)}) = \mathrm{rank}(U \Lambda L^\top) = r.$$

By symmetry of $T$, the same argument gives $\mathrm{rank}(T_{(2)}) = \mathrm{rank}(T_{(3)}) = r$. Moreover, from (14) we have $\mathrm{rank}_{\mathrm{CP}}(T) \leq r$. Since every flattening rank is a lower bound on CP rank (Christandl & Zuiddam, 2019, Section 2.3), it follows that $\mathrm{rank}_{\mathrm{CP}}(T) = r$.

We now use these rank constraints to recover the missing entries of $M_3$. Fix a restriction of size $m$ and let $F := T_{(1)}$ denote the mode-1 unfolding, viewed as a matrix of size $m \times m^2$. Index the columns of $F$ by ordered pairs $(j, k) \in [m] \times [m]$, so that

$$F_{i,(j,k)} = T_{i,j,k}.$$

From the above discussion, in the generic setting we have $\mathrm{rank}(F) = r$, and hence every $(r + 1) \times (r + 1)$ minor of $F$ vanishes.

---

[2]For example, taking $U$ to be a Vandermonde-type matrix yields $\mathrm{rank}(L) = r$, and hence $\mathrm{rank}(L) = r$ holds generically.

**Recovering entries of the form $T_{i,i,k}$ with $i \neq k$.**  We choose an $(r+1) \times (r+1)$ submatrix of $F$ that contains *exactly one* unknown entry of this form. Concretely, pick index sets

$$R \subseteq [m] \text{ with } |R| = r+1, \qquad C \subseteq [m] \times [m] \text{ with } |C| = r+1,$$

such that: (i) for all but one column $(j,k) \in C$, we have $j, k$ distinct from every $i \in R$, so that the corresponding entries $T_{i,j,k}$ are already known from interpolation (pairwise distinct indices), and (ii) there is exactly one special column $(i^\star, k^\star) \in C$ with $i^\star \in R$ and $k^\star \notin R$, for which the unique unknown entry is $T_{i^\star, i^\star, k^\star}$ (two indices equal). Let $B$ denote the resulting $(r+1) \times (r+1)$ submatrix of $F$ with rows $R$ and columns $C$. Then $\det(B) = 0$ since $\mathrm{rank}(F) = r$. Moreover, $\det(B)$ is *linear* in the single unknown entry $T_{i^\star, i^\star, k^\star}$ (all other entries are known), and expanding along the position of this unknown gives an equation of the form

$$T_{i^\star, i^\star, k^\star} \cdot C_1 + C_2 = 0,$$

where $C_1$ is an $r \times r$ minor of $F$ and $C_2$ depends only on known entries. By a similar argument as in Section E.1 (applied to the rank-$r$ matrix $F$) and accounting for precision as above (here each entry of $T$ can have precision upto $\epsilon^3$), we have $C_1 \neq 0$ generically, and hence we can solve for $T_{i^\star, i^\star, k^\star}$ exactly.

There are $\Theta(m^2)$ such entries (up to symmetry), so all entries with exactly two equal indices can be recovered in time $\tilde{O}(m^2 r^{1+\omega})$, in the oracle access model.

**Recovering entries of the form $T_{i,i,i}$.**  Similarly, we recover diagonal entries $T_{i,i,i}$ by selecting a minor containing exactly one entry with all three indices equal, while all other entries correspond to pairwise distinct indices. Since there are only $m$ such values, this takes time $\tilde{O}(mr^{1+\omega})$ after accounting for precision.

Putting the two cases together, the total time (in the oracle access model) to reconstruct all missing entries of $M_3$ exactly is

$$\tilde{O}(m^2 r^{1+\omega}).$$

Adding the time $O(m^3)$ required to recover all entries with pairwise distinct indices via interpolation, we can recover the full *exact* tensor $M_3$ (in the oracle access model) in time

$$\tilde{O}(m^3 + m^2 r^{1+\omega}).$$

and in time $\tilde{O}(sm^3 + m^2 r^{1+\omega})$ in the general setting. Further, the query complexity was only $O(m^3)$.

Combining with the discussion on recovering $M_2$, the total query complexity is $O(m^3)$ for each alive set. The running time is: $\tilde{O}(m^3 + m^2 r^{1+\omega})$ in the oracle access model, and $\tilde{O}(sm^3 + m^2 r^{1+\omega})$ in the general case. Using Proposition 3.1, we only need $m \geq r+1$ to preserve the fan-in $r$. Further, to reconstruct all missing entries of $M_2$ and $M_3$ using the above minor-vanishing procedure, it suffices to take $m \geq 3r + 2$. Thus, we can run our algorithm with restrictions of size $m = 3r + 2$. Since $m = O(r)$, this gives a running time of $\tilde{O}(r^{3+\omega})$ (in the oracle access model) for recovering $M_2$ and $M_3$ exactly on a restriction.

*Remark* E.2. Our main speedup comes from exploiting Proposition 3.1. If we apply the method of moments naively on $n$ variables, then reconstructing $M_2$ and $M_3$ exactly would take $O(n^3)$ queries and a running time of $\tilde{O}(n^3 + n^2 r^{1+\omega})$ in the oracle access model.

## F. Speeding up the whitening process for $M_3$

We saw that whitening $M_3$ to get $\hat{M}_3$ requires us to do the following computation:

$$(\hat{M}_3)_{abc} = \sum_{i=1}^{m} \sum_{j=1}^{m} \sum_{l=1}^{m} W_{i,a} W_{j,b} W_{l,c} (M_3)_{ijl}$$

Doing the multiplication naively would lead to a running time of $O(m^3 r^3)$. However, using multiplication along the different flattenings, we can achieve significant speed-up in this step. We demonstrate the method now: In the first step, we compute the first intermediate tensor $T_1$ of size $r \times m \times m$ using $M_3$ such that

$$(T_1)_{ajl} = \sum_{i=1}^{m} W_{i,a} (M_3)_{ijl}$$

We have $rm^2$ entries and computing each entry takes $m$ arithmetic operations. Hence, total time to compute $T_1$ is $O(rm^3)$. Once, we have $T_1$, we use it to get $T_2$, a $r \times r \times m$ tensor. Formally,

$$(T_2)_{abl} = \sum_{j=1}^{m} W_{j,b}(T_1)_{ajl}$$

Here we have $r^2 m$ entries, and hence time taken is $O(r^2 m^2)$. Finally, we use $T_2$ to get $\hat{M}_3$.

$$(\hat{M}_3)_{abc} = \sum_{l=1}^{m} W_{l,c}(T_2)_{abl}$$

Here we have $r^3$ entries and hence time is $O(r^3 m)$. Hence total time to get $\hat{M}_3$ from $M_3$ is $O(rm^3 + r^2 m^2 + r^3 m)$.

## G. Uniqueness of projections

For a fixed fan-in $r$, for every alive set $A$ we know the number of variables we need to learn, namely $m := |A|$. Moreover, by Proposition 3.1, as long as $m \geq r + 1$, the projected distribution generically has exactly $r$ mixture components, and hence admits a rank-$r$ moment decomposition for any projection.

However, a priori it is still possible that a particular projection is *not identifiable*: namely, there could exist two different parameter tuples that yield the same projected PGP. Concretely, there may exist $f(x_1, \ldots, x_m) = \sum_{i=1}^{r} \mu_i \prod_{j=1}^{m} (\alpha_{ij} x_j + 1)$, and $g(x_1, \ldots, x_m) = \sum_{i=1}^{r} \delta_i \prod_{j=1}^{m} (\beta_{ij} x_j + 1)$, such that $f = g$ as polynomials, even though $(\mu_i, \alpha_{ij})$ and $(\delta_i, \beta_{ij})$ are not the same up to permutation. Equality of polynomials implies equality of all coefficients, and hence for every $\{k_1, \ldots, k_\ell\} \subseteq [m]$ we must have $\sum_{i=1}^{r} \mu_i \prod_{t=1}^{\ell} \alpha_{i,k_t} = \sum_{i=1}^{r} \delta_i \prod_{t=1}^{\ell} \beta_{i,k_t}$.

We show that such non-uniqueness cannot occur in our (generic) setting: for a generic choice of parameters, the projected moments determine the mixture components uniquely (up to permutation), and therefore the above situation is impossible.

We recall that the *Kruskal rank $k_A$* of a matrix $A$ is the largest integer $k$ such that every set of $k$ columns of $A$ is linearly independent.

**Theorem G.1** (Kruskal uniqueness for tensor decomposition, Kruskal (1977)). *Let $T \in \mathbb{R}^{m \times m \times m}$ admit a rank-$r$ tensor decomposition of the form*

$$T = \sum_{i=1}^{r} a_i \otimes b_i \otimes c_i,$$

*and let $A = [a_1| \cdots |a_r]$, $B = [b_1| \cdots |b_r]$, $C = [c_1| \cdots |c_r]$ be the corresponding factor matrices. Let $k_A, k_B, k_C$ denote their Kruskal ranks. If*

$$k_A + k_B + k_C \geq 2r + 2,$$

*then the above decomposition is unique up to permutation of the $r$ terms and rescaling within each term.*

Since we are looking at tensors of the form Equation 14, it is a special case of Theorem G.1 with factor matrices $A = B = C = U$, where $U = [u_1| \cdots |u_r] \in \mathbb{R}^{m \times r}$. Therefore, uniqueness holds whenever

$$3k_U \geq 2r + 2.$$

For $m \geq r$, the matrix $U$ has Kruskal rank $k_U = r$ for generic choices of $\{u_i\}_{i=1}^{r}$, and hence the decomposition $T = \sum_{i=1}^{r} \mu_i u_i^{\otimes 3}$ is generically unique up to permutation and rescaling. Since we also use the second-order moments for reconstruction, we can always avoid the scaling issue simply as the method of moments learns the correct weights $\mu_i$ and parameters $\alpha_{ij}$ from given $M_2, M_3$.

## H. Avoiding collisions in reconstruction

We would first like to point out that suppose we look at a projection under an alive set $A$ as shown in Equation 2. Then, for distinct $\mu_i$, the $\{\mu_i\}$ themselves act as the identifier of the particular mixture. However, suppose w.l.o.g. $\mu_1 = \mu_2$. Then, it can't be the case that $\prod_{j \in A}(\alpha_{1j} x_j + 1) = \prod_{j \in A}(\alpha_{2j} x_j + 1)$, as then in the particular projection $A$, the fan-in would not be $r$, which contradicts Proposition 3.1.

**Proposition H.1.** *Suppose $\mu_i = \mu_j$. For projections of size $m \geq r + 1$, the corresponding product distributions can't be identical in the "generic" setting.*

However, we can still run into the following problem: Suppose, our original distribution was parametrized as following: We have (a) $r = 2, n = 4, \mu_1 = \mu_2 = 0.5$, (b) $(\alpha_{11}, \alpha_{12}, \alpha_{13}, \alpha_{14}) = (0.1, 0.2, 0.3, 0.4)$, and (c) $(\alpha_{21}, \alpha_{22}, \alpha_{23}, \alpha_{24}) = (0.6, 0.7, 0.8, 0.9)$.

Let our first alive set be $A_1 = \{1, 2\}$ and second alive set be $A_2 = \{3, 4\}$. Using our reconstruction algorithm with the restriction to $A_1$ and ignoring permutations, we will recover $\mu_1 = \mu_2 = 0.5$, and $(\alpha_{11}, \alpha_{12}) = (0.1, 0.2)$ and $(\alpha_{21}, \alpha_{22}) = (0.6, 0.7)$. However, when we now take the alive set $A_2$, since we only look at $f = \sum_{i=1}^{r} \mu_i \prod_{j \in A_2}(\alpha_{ij}x_j + 1)$ and $\mu_1 = \mu_2$, we would have no way to distinguish whether $(\alpha_{13}, \alpha_{14})$ takes the values $(0.3, 0.4)$ or $(0.8, 0.9)$. Clearly, if we choose the second tuple, then we will have recovered an incorrect distribution. However, we now show that by only using an overhead of at most $r(r-1)$ parameters (in fact, one parameter per mixture for $r-1$ distinct variables), we can always avoid the above error. For instance, in the above example, we first set the alive set $A_1 = \{1, 2\}$. But now, instead of using $A_2 = \{3, 4\}$, we will use $A_2 = \{1, 3, 4\}$ (equivalently, $\{2, 3, 4\}$). Then, the recovered tuples under $A_2$ would be $(0.1, 0.3, 0.4)$ and $(0.6, 0.8, 0.9)$. Since by $A_1$ we already know the mixture the probabilities $0.1$ and $0.6$ correspond to, we can recover the other parameters without any errors. Here, $r$ was 2 and we only needed to carry $r(r-1) = 2$ parameters $0.1, 0.6$. Formally, we have

**Lemma H.2** (Carry overhead is small). *Given $f = \sum_{i=1}^{r} \mu_i \prod_{j=1}^{n}(\alpha_{ij}x_j + 1)$, assuming a correct reconstruction algorithm, we can recover the correct distribution by requiring an overhead of at most $r(r-1)$ parameters (equivalently, one parameter value for each mixture for each of the $r-1$ distinct variables).*

*Proof.* Without loss of generality, suppose there is a subset $S \subseteq [r]$ of size $k$ such that

$$S = \{1, 2, \ldots, k\} \qquad \text{and} \qquad \mu_1 = \mu_2 = \cdots = \mu_k = \mu.$$

Fix an alive set $A$ of size $m \geq r + 1$. By relabeling variables, we may assume $A = [m]$.

We claim that there exists a subset $V \subseteq A$ of size $|V| = k - 1$ such that for every $i \in S$, and for every $\ell \in S \setminus \{i\}$, there exists some $j \in V$ with

$$\alpha_{ij} \neq \alpha_{\ell j}.$$

In other words, by examining the parameters on the coordinates indexed by $V$, we can distinguish all $k$ mixture components in $S$.

For $k = 1$ there is nothing to prove. For $k = 2$, Proposition H.1 implies that the two components must differ on at least one coordinate $j \in A$, i.e., $\alpha_{1j} \neq \alpha_{2j}$. By permuting coordinates, assume $j = 1$. Thus, to distinguish the two mixtures it suffices to retain the single variable $x_1$, and hence $T_2 \leq 1$.

Let $T_k$ denote the minimum number of variables (coordinates) needed to distinguish $k$ mixture components whose weights are all equal (i.e., $\mu_1 = \cdots = \mu_k$). We have $T_1 = 0$ and $T_2 \leq 1$. Assume inductively that

$$T_\ell \leq \ell - 1 \qquad \text{for all } \ell < k.$$

We now prove $T_k \leq k - 1$.

By Proposition H.1, there exists at least one coordinate $b \in A$ such that not all of $\alpha_{1b}, \ldots, \alpha_{kb}$ are equal (otherwise all $k$ components would be identical). By permuting coordinates, assume $b = 1$. Consider the multiset $\{\alpha_{i1}\}_{i \in S}$ and partition $S$ into buckets $B_1, \ldots, B_g$ such that two indices $i, i' \in S$ lie in the same bucket iff $\alpha_{i1} = \alpha_{i'1}$. Then $g \geq 2$ and $\sum_{t=1}^{g} |B_t| = k$.

The coordinate $x_1$ separates different buckets, but does not distinguish components within the same bucket. Therefore, to fully distinguish all $k$ components, it suffices to: (i) keep coordinate $x_1$, and (ii) within each bucket $B_t$, keep $T_{|B_t|}$ additional coordinates to distinguish its members. Hence,

$$T_k \leq 1 + \sum_{t=1}^{g} T_{|B_t|}.$$

Using the inductive hypothesis $T_{|B_t|} \leq |B_t| - 1$, we obtain

$$T_k \;\leq\; 1 + \sum_{t=1}^{g}(|B_t| - 1) \;=\; 1 + \Big(\sum_{t=1}^{g}|B_t|\Big) - g \;=\; 1 + k - g \;\leq\; k - 1,$$

since $g \geq 2$. This proves the claim that $k - 1$ variables suffice to distinguish the $k$ components in $S$.

Finally, in the general case we can partition $[r]$ into $h \geq 1$ buckets $S_1, \ldots, S_h$, where within each $S_j$ all mixture weights are equal. Applying the above argument to each bucket, we can distinguish the components inside $S_j$ using at most $|S_j| - 1$ variables. Therefore, the total number of variables we need to carry over is at most

$$\sum_{j=1}^{h}(|S_j| - 1) \;=\; r - h \;\leq\; r - 1.$$

Since our initial alive set has size $m \geq r + 1$, we can choose a subset $A' \subseteq A$ of size at most $r - 1$ containing these distinguishing coordinates, and thereafter ensure that every subsequent alive set contains $A'$. This guarantees that across projections we never lose the ability to distinguish mixture components, and we can repeat the procedure until we recover all parameters $\alpha_{ij}$ for all $j \in [n]$. $\qquad\square$

Recall that to properly reconstruct the third-order moments we needed $m \geq 3r + 2$. Now, to ensure we don't confuse between the parameters Lemma H.2 tells us that we need $r - 1$ more variables. Thus, with $m = (3r+2) + (r-1) = 4r+1$, we can always recover the parameters without any issues.

