# OpenReview forum: "Fast Reconstruction of Mixtures of Bernoulli Product Distributions"
_ICML.cc/2026/Conference — ICML 2026 regular_

### Official Review · Reviewer_wz8H · 2026-03-08

**Soundness:** 3
**Presentation:** 3
**Significance:** 3
**Originality:** 3
**Overall Recommendation:** 4
**Confidence:** 3

**Summary:**

This work studies the problem of __reconstructing mixtures of Bernoulli product distributions__ , which means recovering the parameters of $r$ product distributions.

They assume that they have access to its __probability generating polynomial__ (PGP), and they prove that under a certain genericity assumption which excludes some non-identifying cases, that their algorithm suceeds to find those parameters with a __query complexity__ which is $O(nr)$, and a __computational complexity__ given by $O(nr^{4+\delta})$ up to log factors, and errors factors.

Their algorithm proceeds in two steps:
* Step 1: They preprocess the data, and use some genericity results to show that they can restrict to a lower instance such that this restriction characterizes the initial problem. They reduce the problem to determine the principal components of a low-rank tensor
* Step 2: Compute the moments and recover parameters from these moments.

**Compliance With Llm Reviewing Policy:**

Affirmed.

**Final Justification:**

I believe my concerns have been adequately addressed by the rebuttal. I keep my score.

**Key Questions For Authors:**

* The paper assumes oracle access to the probability generating polynomial (PGP) (or an equivalent circuit/graph representation that permits evaluation at non-Boolean inputs). Can the authors clarify in what concrete settings this access is reasonable? Are there known guarantees (even in restricted families) that one can learn/compile such a PGP/PGC graph efficiently?

* __Stability__: Can the authors give quantitative (non-generic) conditions ensuring the cofactors used in the vanishing-minor step are bounded away from zero, or provide a robust variant that tolerates bounded oracle noise?

* The method leverages the algebraic structure of the PGP and moment/tensor reconstruction for mixtures of Bernoulli product distributions. Do the authors expect any part of the pipeline to extend to continuous families, e.g. mixtures of (diagonal/product) Gaussians or more general exponential-family products?

**Limitations:**

yes

**Strengths And Weaknesses:**

## Strengths
* The paper is well-written and clear.
* __Main result:__ exact recovery with $O(nr^{2})$ queries in the generic setting
* __Technically nice ingredient:__ recovering missing diagonal / repeated-index moment entries using rank constraints and vanishing minors (instead of matrix completion / least squares). This is plausibly of independent intere


## Weakness

* __Strong / nonstandard access model.__ The algorithm assumes oracle access to the probability generating polynomial (or an equivalent circuit) and evaluates it on non-Boolean inputs. This is substantially stronger than the usual sample-based learning model, so the practical scope of the result is unclear unless the paper explains in which realistic scenarios such oracle access is available and why recovering mixture parameters is needed once a circuit representation is known.

* __Genericity vs robustness.__ Several steps rely on “generic” non-degeneracy assumptions (e.g., certain minors/cofactors being nonzero). Even if these hold with probability 1 under random parameters, they may be poorly conditioned (very small cofactors), which raises concerns about numerical stability and the meaning of “exact recovery” in finite precision. The paper would benefit from a conditioning discussion or a robust/noisy extension.

---

> ### Author Rebuttal · Authors · 2026-03-30
>
> Thank you for your feedback. We will like to present the following responses to the points you raised in your review.
>
> 1. Our algorithm is fundamentally about recovering distribution parameters using query access given a useful representation learned from samples. Often times though, these representations can be very abstract and give no access to these parameters. For a detailed overview, please refer to the answer for Reviewer qhG2 under the heading "Learning Probabilistic circuits" and the answer to Reviewer rzHg under the heading "Representation as a PGC". This is essential as recovery of parameters helps us in answering all sorts of inference queries which let us reason about the data. There is already a strong line of work [1,2,3,4] studying such inference queries like marginalization, maximum a posteriori (MAP) inference, marginal MAP, among others. We therefore view our work as solving a relevant problem for the community: Having learned a compact representation, when can we efficiently recover the model parameters from query access?
>
> 2. Thank you for your question. The issues you have mentioned appear frequently in finite precision reconstruction. For the co-factors being close to $0$, we implicitly operate in the setting where the parameters have a finite precision $\zeta$. In this setting, any restriction matrix has size $O(r)\times O(r)$, and hence the precision of the determinants is atmost $O(r^2\zeta)$. Thus, we only need to evaluate upto $O(\log r + \log \zeta)$ bits, which only adds a log factor overhead to the algorithm. Since we use the algorithm of [5] as a black box we will also refer you to the discussion on robustness in [5]. Further, we will request you to see our answer to reviewer EkNn under the heading "Precision of the algorithm".
>
> 3. Once we look at a continuous distribution, it is not even clear what a probability generating polynomial looks like. We haven't thought about this question in detail yet, but intuitively, we don't expect our methods to translate to such settings.
>
> [1] \textit{Bl\"aser}. Not all strongly Rayleigh distributions have small probabilistic generating circuits. ICML 2023.
>
> [2] \textit{Broadrick et al}. The Limits of Tractable Marginalization. ICML 2025.
>
> [3] \textit{Mei et al}. Maximum A Posteriori Inference in Sum-Product Networks. AAAI 2018.
>
> [4] \textit{de Campos}. New complexity results for map in bayesian networks. IJCAI 2011.
>
> [5] \textit{Anandkumar et. al}. Tensor Decompositions for Learning Latent Variable Models. JMLR 2014.

---

> > ### Author Rebuttal · Reviewer_wz8H · 2026-04-01
> >
> > Thank you to the authors for your thoughtful responses.

---

> > > ### Author Response · Authors · 2026-04-04
> > >
> > > We thank the reviewer for the positive acknowledgment of our answers. If there are no remaining concerns, we kindly ask you to consider updating your score to reflect that the issues have been fully addressed.

---

### Official Review · Reviewer_qhG2 · 2026-03-11

**Soundness:** 2
**Presentation:** 3
**Significance:** 2
**Originality:** 3
**Overall Recommendation:** 3
**Confidence:** 5

**Summary:**

The paper provides an algorithm to recover the parameters of the Probability Generating Polynomial (PGP) of mixtures of binary product distributions. The algorithm makes the recovery in the number of queries, which are linear in the number of random variables/samples and quadratic in the number of mixtures of product distributions.

**Compliance With Llm Reviewing Policy:**

Affirmed.

**Ethical Review Concerns:**

yes

**Final Justification:**

Thanks to the authors for their rebuttal, I will consider increasing the score by 1 point. However, considering the scope, limitations of the current work and nature of the problem being addressed in the paper, my general review of the paper remains almost the same.

**Key Questions For Authors:**

1. Can the algorithm be extended to the case of ternary distributions or further with a worse query complexity, perhaps?

2.  The algorithm assumes that the PGP for the mixture is known. How strict is such an assumption? In most application settings, I would believe that one may have access to data points sampled from the mixture but not PGP in general, unless PGP is estimated from the data points available to the user?

3. I think there is a typo/error in Figure 1 in the table.

4. Can authors show the applicability of this algorithm to a related application and how it outperforms other currently used methods to recover 'Bernoulli' marginal distributions in terms of query complexity or running time?

**Limitations:**

yes

**Strengths And Weaknesses:**

Strengths:

1. The paper provides a novel algorithm for recovering the parameters of the mixtures and marginal distributions from the PGP of a mixture of Bernoulli product distributions with linear query complexity in the number of variables and quadratic in the number of mixtures.

2. I have gone through the main content of the paper, and the analysis and intuition behind the algorithm looks fine to me. However, I have not thoroughly verified the proofs in the appendix.

Weaknesses:

1. I don't see why the result presented in this paper aligns with the interests of this community. I believe that such a theoretical result should be presented in a TCS or probability community. It would be nice if authors can shed more light on this.

2. The only algorithm presented in the paper works 'specifically' for mixtures of 'Bernoulli' product distributions.

3. The paper mentions in a few lines that a mixture of product distributions shows up in a lot of applications, such as image recognition, medical imaging, crowdsourcing, etc., but none of the applications have been discussed in detail, nor has been mentioned about the applicability of the algorithm presented in this paper

---

> ### Author Rebuttal · Authors · 2026-03-30
>
> Thank you for your feedback. We will like to present the following responses to the points you raised in your review.
>
> \textbf{Relevance of these models}: We agree that the paper is theoretical in nature, but we believe it is well aligned with ICML because it studies tractable probabilistic representations for high-dimensional data, an established theme in machine learning community [4,5,7,8] and at ICML in particular [1,3,11]. Beyond their intrinsic theoretical interest, these models have also been used in a growing range of applications, including exact inference, probabilistic reasoning, and more recently control and alignment of deep generative models [2,10], among others [6,9]. We therefore view our result as relevant to ICML: it gives the fastest known algorithm for recovering such representations and thereby reasoning explicitly about the underlying distribution.
>
> \textbf{Universality of Bernoulli mixtures}: Indeed our algorithm only works for Bernoulli mixtures, however they are one of the most general ways to represent heterogeneity in high-dimensional binary data: conditioned on a latent class, coordinates are independent, and overall population is a mixture across classes. Naturally, such models are widely prevalent across statistics and ML [12]. In particular, many practical problems reduce to this form after binarization (e.g., binary features/attributes, presence–absence indicators, or one-hot encodings). In fact, it is known that any discrete distribution can be represented as a mixture of product components [13].
>
> An immediate application of our algorithm is enhancing the tractable methods for PGCs and PCs. Figure 2 in [11] describe an architecture how to learn distributions of binary variables and store them in a PGC. This PGC architecture can contains DPPs as substructures and therefore, the parameters of the mixture cannot be read off from the PGC. Our methods reconstructs these parameters significantly faster than state-off the art reconstructions methods ($O(nr^2)$ compared to $O(n 2^{r^2})$).
>
> Answering the key questions you raised:
> 1. In general, if ternary distributions are represented using non-multilinear polynomials, then PGCs are known to lose tractability [1]. Another possible approach is to represent each ternary variable using three indicator variables while preserving multilinearity. We have considered this viewpoint, but at present it is not clear whether our reconstruction method extends to that setting in a direct way.
>
> 2. \textbf{Learning Probabilistic Circuits}: There has been substantial recent progress on improving both the learnability and scalability of such circuits, and practical algorithms for learning them from data are now available [3,14,15]. A key advantage of these models is that they support exact inference queries efficiently, which makes them particularly attractive in applications where reliable probabilistic reasoning is important.
>
> 3. Yes, thanks for catching this. The row and column labels in Figure 1 should indeed be $X=0, X=1$ and $Y=0, Y=1$, respectively.
>
> 4. As discussed in Sections 1.2 and 1.3, our algorithm achieves the best known query complexity, namely $O(nr^2)$, under the assumptions considered in our paper. By contrast, the best previous source-identification algorithms [16] require query complexity $n2^{r^2}$, which is exponential in $r$. Our improvement comes from exploiting the structure of the reconstruction problem more directly, thereby avoiding the exponential dependence on $r$ present in prior approaches.
>
> [1] Agarwal et al. Probabilistic Generating Circuits - Demystified. ICML 2024.
>
> [2] Liu et al. Image Inpainting via Tractable Steering of Diffusion Models. ICLR 2024.
>
> [3] Liu et al. Scaling Tractable Probabilistic Circuits: A Systems Perspective. ICML 2024.
>
> [4] Loconte et al. Sum of Squares Circuits. AAAI 2025.
>
> [5] Martens et al. On the expressive-efficiency of sum-product networks. NeurIPS 2013.
>
> [6] Saad et al. SPPL: Probabilistic Programming with Fast Exact Symbolic Inference. PLDI 2021.
>
> [7] Vergari et al. A Compositional Atlas of Tractable Circuit Operations for Probabilistic Inference. NeurIPS 2021.
>
> [8] Wang et al. A Compositional Atlas of Algebraic Circuits. NeurIPS 2024.
>
> [9] Wedenig et al. Exact Soft Analytical Side-Channel Attacks using Tractable Circuits. ICML 2024.
>
> [10] Zhang et al. Adaptable Logical Control for Large Language Models. NeurIPS 2024.
>
> [11] Zhang et al. Probabilistic Generating Circuits. ICML 2021.
>
> [12] McLachlan \& Peel. Finite Mixture Models. 2000.
>
> [13] J. Grim. EM cluster analysis for categorical data.  Structural, Syntactic, and Statistical Pattern Recognition, 2006.
>
> [14] Zhang et al. Scaling Probabilistic Circuits via Monarch Matrices. ICML 2025.
>
> [15] Gala et al. Scaling Continuous Latent Variable Models as Probabilistic Integral Circuits. NeurIPS 2024.
>
> [16] Gordon et al. Source identification for mixtures of product distributions. COLT 2021.

---

> > ### Author Rebuttal · Reviewer_qhG2 · 2026-04-03
> >
> > I would like to thank the reviewers for their responses.
> >
> > For the response to Key Question 2: My concern is about the practical availability of the mixture's PGP, as assumed in the work. The response does not seem to address this directly. In practical settings, I would like to know why it is reasonable to assume that PGP is known.  If it is unknown and must be estimated from the samples, how would this affect the algorithm and the guarantees in the paper, specifically the recovery error and the computational complexity?

---

> > > ### Author Response · Authors · 2026-04-04
> > >
> > > Thanks for you helpful comments. As you mention, we only have samples available initially. What we wanted to highlight in our answer was the following: Given samples, we can effectively learn a representation of the PGP of the distribution. Once done, we can then reconstruct the parameters using our algorithm.
> > >
> > > In our main application scenario, we assume that we already have learned a PGC, storing the PGP, from data. Zhang et. al [1] for instance provide such an architecture, as well as [2,3,4]. Once the PGC is learned, we consider the stored distribution to be the ground truth. Therefore, in this work, we do not focus on the error rate of the learning process as well as its time complexity. To enhance expressiveness, PGCs might store distributions not as mixtures of product distributions but in a completely different way, for instance, using DPPs as subcircuits. Our algorithm can be viewed as an enhancement of the tractable methods that are available for PCs and PGCs, see [5]. Instead of sample-based learning, our work should be considered as query-based reconstruction. While PGCs store distributions differently, by the universality of mixtures of Bernoullis, the stored distribution can be written as such a mixture of Bernoullis. Our algorithms allows the fastest known extraction of the parameters of the mixture from the learned circuit. By having query access, we can do this much faster than sample-based learning.
> > >
> > > We hope that this sufficiently answers your queries. If that is the case, we would kindly request you to consider updating the score to reflect that.
> > >
> > > $[1]$ \textit{Zhang et al.} Probabilistic Generating Circuits. ICML 2021.
> > >
> > > $\lbrack2\rbrack$ \textit{Liu et al.} Scaling Tractable Probabilistic Circuits: A Systems Perspective. ICML 2024.
> > >
> > > $\lbrack3\rbrack$ \textit{Zhang et al.} Scaling Probabilistic Circuits via Monarch Matrices. ICML 2025.
> > >
> > > $\lbrack4\rbrack$ \textit{Gala et al.} Scaling Continuous Latent Variable Models as Probabilistic Integral Circuits. NeurIPS 2024.
> > >
> > > $[5]$ \textit{Vergari et al.} A Compositional Atlas of Tractable Circuit Operations for Probabilistic Inference. NeurIPS 2021.

---

### Official Review · Reviewer_EkNn · 2026-03-13

**Soundness:** 4
**Presentation:** 3
**Significance:** 2
**Originality:** 4
**Overall Recommendation:** 4
**Confidence:** 2

**Summary:**

This paper studies reconstruction of an $r$-component mixture of Bernoulli product distributions from oracle access to its probability generating polynomial (PGP). The main result is an exact reconstruction algorithm using $O(nr^2)$ oracle queries. This holds in a "generic setting", i.e. that reconstruction succeeds for all parameter choices outside a proper algebraic exceptional (measure-zero) set, so the guarantee is almost-everywhere rather than worst-case. This is the first such result with linear dependence on the ambient dimension $n$ in this access model.

Methodologically, the algorithm first preprocess the PGP to remove irrelevant variables and common linear factors. Then it repeatedly restrict the mixture to only $O(r)$ variables and show that such restrictions preserve the original mixture size $r$. On each restricted instance, the algorithm recovers low-order moments from the resulting coefficients, fills in the missing diagonal moment entries from rank constraints, and then applies a standard tensor-decomposition to recover the component parameters (up to permutatons).  Finally, it stitches these restricted solutions together to obtain the full $n$-dimensional model.

**Compliance With Llm Reviewing Policy:**

Affirmed.

**Final Justification:**

I don't have major questions for the paper to begin with (and the most significant one -- the appearance of an accuracy parameter in the "exact reconstruction" algorithm -- is properly clarified as due to the bit precision issue.) Therefore, I will keep my positive score unchanged.

**Key Questions For Authors:**

See Weakness section.

**Limitations:**

Yes.

**Strengths And Weaknesses:**

**Strength**

- As a theoretical paper, I think this paper is technically strong and reasonably original. The main contribution is not simply another use of moment / tensor methods, but a fairly well-executed framework that decomposes the $n$-dimensional reconstruction problem into $O(r)$-variable restrictions, with careful proofs that such restrictions generically preserve the mixture size and that one can reconstruct the missing diagonal moment and then stitches solutions back together. This overall strategy feels conceptually clean and nontrivial.

- The result itself is also strong within the stated model: the paper claims exact recovery in the PGP oracle model using $O(nr^2)$ queries. This linear dependence on the ambient dimension appears to compare favorably to prior works discussed in the paper that either target approximate recovery or have substantially worse dependence on $n$ and $r$. (I think it is especially interesting to obtain *exact* parameter recovery rather than only approximate recovery from samples. So even if the model is specialized, this feels like a meaningful theoretical result.)


**Weakness**

- My main reservation is about significance and practical relevance. The access model is quite specialized, and the motivating application through PGCs felt weaker than the paper suggests. The algorithm itself only assumes black-box query access to the PGP and does not exploit any circuit structure, so the repeated discussion of PGCs did not seem very relevant to me, especially for an ICML audience. Put differently, the theory result is interesting on its own terms, but the paper's ML motivation feels somewhat indirect.

- I think the paper could be a bit clearer about what exactly is meant computationally by "exact reconstruction", since the runtime discussion later seems to rely on a tensor-power routine parameterized by a target accuracy $\varepsilon$. My guess is that this is resolvable, but the relationship between numerical precision and exact recovery could be stated more explicitly. This is a bit confusing especially since "exactness" feels like one of the main significances of the result.

- Some minor things:

(1) I think it would be nice to mention explicitly in Theorem 1.2 that the algorithm takes $r$ as input. Otherwise, the subsequent Remark 1.5 can read a bit contradictory to the theorem.

(2) (If I read correctly) the authors use "generic setting" a couple of times to describe that reconstruction succeeds for all parameter choices outside a proper algebraic exceptional (measure-zero) set, so the guarantee is almost-everywhere rather than worst-case. I do not view this as a flaw, but this certainly narrows the scope of the result rather than making it more "generic", so I suggest change this wording.

---

> ### Author Rebuttal · Authors · 2026-03-30
>
> Thank you for your feedback. We will like to present the following responses to the points you raised in your review.
>
> First, we would like to point out that there has been extensive recent work on such probabilistic circuit models, and they are particularly useful for tasks requiring exact inference and reasoning. For a detailed answer, see our reply to Reviewer qhG2 on "Relevance of such models". It is also known that all these probabilistic representations are interchangeable into one another in polynomial time in the binary setting [1]. Hence, we focus on PGCs/PCs as there has been a lot of work on practical algorithms for learning such representations, see also the answer to Reviewer qhG2 on "Learning Probabilistic circuits".
>
> \textbf{Precision of the algorithm}: This is a standard issue in finite-precision reconstruction, which we had left implicit in the paper. As in prior work, we assume that the parameters of the underlying distribution are specified up to some fixed precision $\zeta$. Indeed, if the parameters were arbitrary irrationals, then no practical algorithm could hope to recover them with infinite precision. Under this assumption, our algorithm recovers the parameters up to $\zeta$-accuracy, with convergence in $O(\log \zeta)$ iterations. Such a precision/separation assumption is standard in source-identification results: for example, [2] shows that source identification is possible if and only if at least $2r-1$ random variables have mixture probabilities that are mutually distinct and separated by $\zeta$. In this case, [2] give a $n^{O(r)}2^{r^2}$ query complexity algorithm for identification. Further, if they have all RVs $\zeta$-separated then they can give a $n2^{r^2}$ query complexity algorithm. In the same context, if we have all random variables as $\zeta$-separated our algorithm will return the parameters upto $\zeta$ precision in $O(nr^2)$ query complexity. Also importantly, even when only $2r-1$ random variables are $\zeta$-separated, our method performs substantially better than [2]. An implication of Lemma H.2 in our paper is that if one fixed variable, say $X_1$, is known to be $\zeta$-separated, then keeping $X_1$ alive in every restriction already suffices to distinguish all mixture components. The main challenge is that we do not know beforehand which variables are the $\zeta$-separated ones. In [2], this necessitates a brute-force search over all candidate subsets of size $2r-1$, giving rise to the $n^{O(r)}$ overhead. By contrast, our algorithm needs only \emph{one} good variable! So, it suffices to brute-force over the $n$ possible choices of that variable. As a result, when only $2r-1$ variables are $\zeta$-separated, we obtain query complexity $O(n^2r^2)$, significantly improving over the complexity in [2]. We will make this point more explicit in the final submission.
>
> We will now try to address the minor points you mentioned:
>
> 1. Our algorithm does not need $r$ as an input. The main idea is that in the ``generic'' setting, $r$  can in fact be computed in polynomial time, for instance via our algorithm itself, whereas the hardness results concern the worst-case setting. We agree that the current wording makes this distinction insufficiently clear, and we will fix it.
> 2. By ``generic,'' we mean almost-everywhere, not worst-case. We will revise the wording throughout to make this precise. Thank you for pointing this out.
>
>
> [1] \textit{Broadrick et al}. Polynomial Semantics of Tractable Probabilistic Circuits. UAI 2024.
>
> [2] \textit{Gordon et al.} Source identification for mixtures of product distributions. COLT 2021.

---

> > ### Author Rebuttal · Reviewer_EkNn · 2026-04-03
> >
> > Thank you for responding to my questions.
> >
> > Some remarks: I understand that these representations are interchangeable, so my point was that, since the algorithm only assumes black-box query access to the PGP, perhaps the authors do not need to emphasize the example of generating circuits so much. Repeated discussion of PGCs can be a bit distracting, as it may lead readers to focus on circuit structures. Also, I suggest including the precision discussion in the paper, or at least being more explicit that the accuracy parameter appears only because of bit-precision issues.

---

> > > ### Author Response · Authors · 2026-04-04
> > >
> > > Thanks for your helpful comments. We will definitely include the discussion about precision in the final submission. We focused on PGCs because it provides an efficient way to learn a representation of the PGP of the distribution from the samples, as mentioned in our response to Reviewer qhG2 under "Learning such circuits". However, we will make this more explicit in the final draft. If this sufficiently answers your concerns, we will kindly request you to consider updating the score to reflect that.

---

### Official Review · Reviewer_rzHG · 2026-03-16

**Soundness:** 3
**Presentation:** 3
**Significance:** 3
**Originality:** 3
**Overall Recommendation:** 3
**Confidence:** 2

**Summary:**

This paper considers the problem of reconstruction of mixtures of Bernoulli product distributions. Specifically, given access to a learned probability generating polynomial (PGP) representation of the distribution mixture, the proposed algorithms recover the parameters of the distribution mixture within $O(nr^2)$ queries to the PGP and running in time $O(nr^{4+\delta}(\log r + \log\log \frac{1}{\epsilon}))$. If considering also the computations for learning the PGP, then the running time requires $O(nsr^2)$ more, where $s$ is the size of the representation of the PGP.

The challenge lies in that the structure of PGP is unknown to the algorithm and it is assumed that the algorithm only has query access to the PGP/PGC. As a result, the authors design algorithms such that the $n$ variables are divided into blocks of size $m=O(r)$, where exact recovery is possible from low-order coefficients through spectral methods. To speed up, here the authors avoid using matrix completion or least-squares optimization, but instead exploit the structure of the moment matrix whose rank is at most $r$. Using such structures, the algorithm is able to construct essentially linear terms of a matrix entry so as to solve them in closed form.

**Compliance With Llm Reviewing Policy:**

Affirmed.

**Key Questions For Authors:**

Line 110: does this mean that PGC can store the PCP in a way that is totally different from what is defined in Eq. (1)? Is there any explanation for this phenomenon?

Line 193: while here is claimed that the algorithm is not able to work on $n$ variables at once, what is the reason of this restriction? Is it computational or query-wise restrictions?

Line 204: while it is explained through proof that the actual effective fan-in is no less than $r$, what exactly will happen when it is smaller than $r$? Will the coefficients degenerated or disappear? I didn't quite get the challenge here.

**Limitations:**

yes

**Strengths And Weaknesses:**

The estimation of distribution parameters of mixture of Bernoulli product distributions is a challenging problem while being useful to many applications. This paper reviews this problem from a algebraic/symbolic point of view. More specifically, while prior works usually rely on sample access, this paper considers having access to the probability generating polynomial, which can be seen as a compact representation of the distribution. The paper discussed many related works and distinguish its approaches from the others.

The approaches is spectral-based, with moment extraction and parameter recovery from the moment information. The algorithm first divides variables into small blocks, performs the spectral method, and then stitches them together for a complete reconstruction. As for now, I still didn't get the point of the "divide-and-then-stitch" strategy.

---

> ### Author Rebuttal · Authors · 2026-03-30
>
> Thank you for your feedback. We will like to present the following responses to the points you raised in your review.
>
> 1. "Divide-and-stitch" is necessary to keep the complexity linear in $n$. Since our method uses moments up to order three, applying it directly to all $n$ variables would require $O(n^3)$ queries. Instead, we partition the variables into $O(n/r)$ blocks of size $O(r)$, process each block separately in $O(r^3)$ time, and then stitch the local reconstructions together, yielding overall complexity $O(nr^2)$.
>
> 2.  \textbf{Representation as a PGC}: Although the PGP ultimately represents the polynomial in Equation (1), the corresponding PGC need not have the explicit form of a mixture of Bernoulli products. A PGC may exploit cancellations or other algebraic structure, leading to representations that look quite different syntactically from a simple sum of products. For example, Zhang et al. [1] give a PGC architecture for learning joint distributions of binary variables that makes use of DPP--based structure, and hence may involve determinants (see also Figure 2 in [1]). Such a learned PGC therefore need not resemble a mixture-of-products representation, even though the underlying binary distribution can still be written in that form. This is because a mixture of Bernoulli products is known to be universal for binary distributions, see also our answer to Reviewer qhG2 under the heading "Universality of Bernoulli mixtures". Our algorithm provides a way to reconstruct such a mixture representation efficiently, and does so faster than previously known approaches.
> $[1]$ \textit{Zhang et al.} Probabilistic Generating Circuits. ICML 2021.
>
> 3.  As you mention, the reason is that it will be computationally expensive to work with all $n$ variables as we need the third order moments which lead to a cubic overhead.
>
> 4. It can happen that the coefficients actually become the same on a restriction and hence the number of summands reduces. For example, take the following PGF of a distribution:
>    $ f(x,y,z) = 0.3(0.3x + 0.7)(0.2y + 0.8)(0.4z + 0.6) + 0.7(0.3x+0.7)(0.4y+0.6)(0.9z+0.1)$
>
>     In this case number of variables $n=3$ and number of summands $r=2$. However if we were to look at the restriction where we marginalize out $y$ and $z$ and keep only $x$ alive, then the new polynomial is $ f(x,1,1) = 0.3x + 0.7$
>
>     Here, the $r$ now reduces to $1$. Hence, a restriction naively can lead to number of summands dropping. Which is what we want to avoid as in this case we will not be able to recover all parameters related to that particular restriction.

---

> > ### Author Rebuttal · Reviewer_rzHG · 2026-04-03
> >
> > The response addressed my questions on why to use the "divide-and-stitch" approach (for computational efficiency) and the issues of forcing restrictions.

---

> > > ### Author Response · Authors · 2026-04-04
> > >
> > > We thank the reviewer for the positive acknowledgment of our answers. If there are no remaining concerns, we kindly ask you to consider updating your score to reflect that the issues have been fully addressed.

---

### Decision · Program_Chairs · 2026-04-30

**Decision:**

Accept (regular)

**Comment:**

This submission studies the problem of reconstructing mixtures of product distributions, which is a classical problem in learning theory. The authors propose an algorithm that solves this problem using $O(n)$ queries. The main concern raised by reviewers is the reliance on a PGP oracle, which may not be available in practice. At the same time, some reviewers acknowledged the significance of this result even under this assumption, and noted that the approach goes beyond yet another application of method-of-moments. This was a borderline case, but the strength of the result combined with the continued relevance of the problem studied makes a strong case for acceptance.

**Note:** One reviewer marked their concerns as "Fully resolved" but did not update their score. This has been taken into account.